# Hippocampus and striatum show distinct contributions to longitudinal changes in value-based learning in middle childhood

**Johannes Falck[1]\*, Lei Zhang[2,3,4,5], Laurel Raffington[6], Johannes Julius Mohn[7,8], Jochen Triesch[9], Christine Heim[7,10], Yee Lee Shing[1]\***

[1]Department of Psychology, Goethe University Frankfurt, Frankfurt, Germany; [2]Centre for Human Brain Health, School of Psychology, University of Birmingham, Birmingham, United Kingdom; [3]Institute for Mental Health, School of Psychology, University of Birmingham, Birmingham, United Kingdom; [4]Centre for Developmental Science, School of Psychology, University of Birmingham, Birmingham, United Kingdom; [5]Social, Cognitive and Affective Neuroscience Unit, Department of Cognition, Emotion, and Methods in Psychology, Faculty of Psychology, University of Vienna, Vienna, Austria; [6]Max Planck Research Group Biosocial, Max Planck Institute for Human Development, Berlin, Germany; [7]Charité – Universitätsmedizin Berlin, Institute of Medical Psychology, Berlin, Germany; [8]Max Planck School of Cognition, Max Planck Institute for Human Cognitive and Brain Sciences, Leipzig, Germany; [9]Frankfurt Institute for Advanced Studies (FIAS), Frankfurt am Main, Germany; [10]Center for Safe & Healthy Children, The Pennsylvania State University, University Park, United States

**\*For correspondence:**
falck@psych.uni-frankfurt.de (JF);
shing@psych.uni-frankfurt.de
(YLS)

**Competing interest:** The authors declare that no competing interests exist.

**Abstract** The hippocampal-dependent memory system and striatal-dependent memory system modulate reinforcement learning depending on feedback timing in adults, but their contributions during development remain unclear. In a 2-year longitudinal study, 6-to-7-year-old children performed a reinforcement learning task in which they received feedback immediately or with a short delay following their response. Children's learning was found to be sensitive to feedback timing modulations in their reaction time and inverse temperature parameter, which quantifies value-guided decision-making. They showed longitudinal improvements towards more optimal value-based learning, and their hippocampal volume showed protracted maturation. Better delayed model-derived learning covaried with larger hippocampal volume longitudinally, in line with the adult literature. In contrast, a larger striatal volume in children was associated with both better immediate and delayed model-derived learning longitudinally. These findings show, for the first time, an early hippocampal contribution to the dynamic development of reinforcement learning in middle childhood, with neurally less differentiated and more cooperative memory systems than in adults.

## eLife assessment

In this work, the authors make a **valuable** contribution based on **convincing** evidence that children 6-to-7-years-old improve in 2 years of development towards utilising more optimal value-based decision-making strategies while performing a reinforcement learning task. They found that delayed feedback learning was associated with volume in the hippocampus while immediate feedback learning was not. Striatal volume was associated with both forms of learning, in contrast to prior research funding in adults. Brain-behaviour correlations were stable across the 2-year period, despite the hippocampus increasing in volume and striatal volume remaining stable.

## Introduction

As children enter school during middle childhood, they must learn to act appropriately in new situations through feedback. For example, children must learn to raise their hand before speaking during class. The teacher may reinforce this behavior immediately or with a delay, which raises the question whether feedback timing modulates their learning. Here, reinforcement learning (RL; *Sutton and Barto, 2018*) provides a useful mechanistic framework to describe such feedback-driven value-based learning and decision-making. RL models allow to explicitly test for the influence of separate components during value-based learning, such as model-free and model-based learning (*Gläscher et al., 2010*), social and non-social learning (*Bolenz et al., 2017*; *Zhang and Gläscher, 2020*), or the contribution of different memory systems (*Foerde and Shohamy, 2011*; *Packard and Goodman, 2013*; *Goodman and Packard, 2016*).

The role of feedback timing has previously been studied in relation to memory sytems. The memory systems account is a theoretical framework that proposes that different types of memory are supported by distinct neural systems in the brain. Specifically, this account suggests that there are two memory systems: a hippocampal-dependent system and a striatal-dependent system. These systems modulate memory and value-based learning, and their interactive development has been of particular interest to developmental research (*Davidow et al., 2016*; *Hartley et al., 2021*). In adults, the hippocampal-dependent memory system has been shown to contribute to episodic memory during reinforcement learning and is more engaged during feedback that is presented with a delay (*Packard and Goodman, 2013*; *Packard et al., 2018*; *Schwabe and Wolf, 2013*), as opposed to the striatal-dependent memory system, which is more engaged after immediate feedback and supports habitual memory (*Foerde and Shohamy, 2011*; *Foerde et al., 2013*; *Höltje and Mecklinger, 2020*; *Lighthall et al., 2018*). Specifically, hippocampal activation was greater during delayed feedback than during immediate feedback, whereas striatal activation was greater during immediate feedback than during delayed feedback (*Foerde and Shohamy, 2011*). The engagement of the hippocampus during delayed feedback was further supported by enhanced episodic memory for incidentally presented objects compared to objects presented with immediate feedback. Taken together, findings from adult studies suggest that feedback timing modulates the engagement of the hippocampal and striatal memory systems during value-based learning. Given the differential developmental trajectories of these systems and the impact the systems have on reinforcement learning and memory, it is important to understand whether children would show similar feedback timing modulations as previously shown in adults. In addition, whether such feedback timing modulation changes over time remains largely unexplored. To this end, in this study, we examined the contributions of hippocampal and striatal structural volumes during the longitudinal development of reinforcement learning across two years in 6-to-7-year-old children. We will introduce the key parameters in reinforcement learning and then we review the existing literature on developmental trajectories in reinforcement learning as well as on hippocampus and striatum, our two brain regions of interest.

Reinforcement learning behavior modulated by feedback timing can be modeled computationally using at least three parameters that reflect feedback-based learning and decision-making. For feedback-based learning, a learning rate parameter determines the extent to which the reward prediction error, defined as the difference between the received reward and the expected reward, influences the update of the future choice values. A higher learning rate emphasizes recent outcomes, whereas a lower learning rate reflects learning integrated over a longer outcome history (*Zhang et al., 2020*). Value updates may further depend on an outcome sensitivity parameter that scales the individual magnitude of received rewards. Finally, in decision-making, the inverse temperature parameter plays a key role in determining the tendency to select the more valuable choice and quantifies choice stochasticity. A higher inverse temperature reflects more value-guided, deterministic choice behavior compared to a lower inverse temperature reflecting more random choices. Learning rates and inverse temperature have been studied extensively across development, mainly with cross-sectional studies showing mixed findings regarding their age gradients (*Nussenbaum and Hartley, 2019*). One study reported lower learning rates in children compared to adolescents (*Decker et al., 2015*), while other studies found no differences (*Javadi et al., 2014*; *Palminteri et al., 2016*) or even higher learning rates in children (*Davidow et al., 2016*; *Master et al., 2020*). Developmental differences regarding the inverse temperature parameter are slightly more consistent, with studies reporting no differences (*Davidow et al., 2016*; *Hauser et al., 2015*; *Moutoussis et al., 2018*; *van den Bos et al., 2012*) or

higher inverse temperature with age that suggests that behavior is increasingly value-guided and less explorative (*Decker et al., 2015*; *Javadi et al., 2014*; *Palminteri et al., 2016*; *Rodriguez Buritica et al., 2019*). To the best of our knowledge, outcome sensitivity has not been modeled computationally across development. However, studies that linked striatal reward activation to self-reported reward sensitivity showed increasing sensitivity from childhood to adolescence (*Galván, 2013*; *van Duijvenvoorde et al., 2014*).

In general, the inconsistencies regarding developmental differences in parameters may be due to their dependency on model and task properties (*Eckstein et al., 2021*), which could be reconciled by comparing developmental changes to simulation-based optimal learning (*Zhang et al., 2020*). Such comparisons acknowledge that optimal parameter values vary depending on the context, and it has been suggested that humans develop towards more optimal parameter values from childhood into adulthood (*Nussenbaum and Hartley, 2019*). Importantly, to our knowledge previous reinforcement learning studies with children were cross-sectional, and only two studies investigated children under 8 years of age (*Decker et al., 2015*; *Cohen et al., 2020*). Cross-sectional studies, in which developmental change is inferred as a between-subject factor, do not capture the dynamics in middle childhood if individual differences are large, whereas longitudinal studies test development as a within-subject factor, which is crucial for uncovering change across time. Thus, longitudinal changes in reinforcement learning in middle childhood as well as their putative striatal and hippocampal associations remain unknown. To this end, learning rates, outcome sensitivity, and inverse temperature are relevant computational parameters to study longitudinal changes in striatal and hippocampal systems during value-based learning.

Striatal and hippocampal contributions to reinforcement learning during middle childhood may differ as these brain regions undergo major developmental changes. Although earlier structural studies with relatively small sample sizes showed large developmental variability and a tendency for an earlier volume peak in the striatum than in the hippocampus (*Raznahan et al., 2014*; *Wierenga et al., 2014*; *Giedd, 2004*; *Uematsu et al., 2012*; *Giedd et al., 2015*; *Goodman et al., 2014*; *Goddings et al., 2014*), a recent cross-sectional large-scale study was able to contrast striatal and hippocampal trajectories with greater granularity (*Dima et al., 2022*). These data showed striatal volume peaks in the first decade which then declined throughout later developmental periods, whereas hippocampal volume

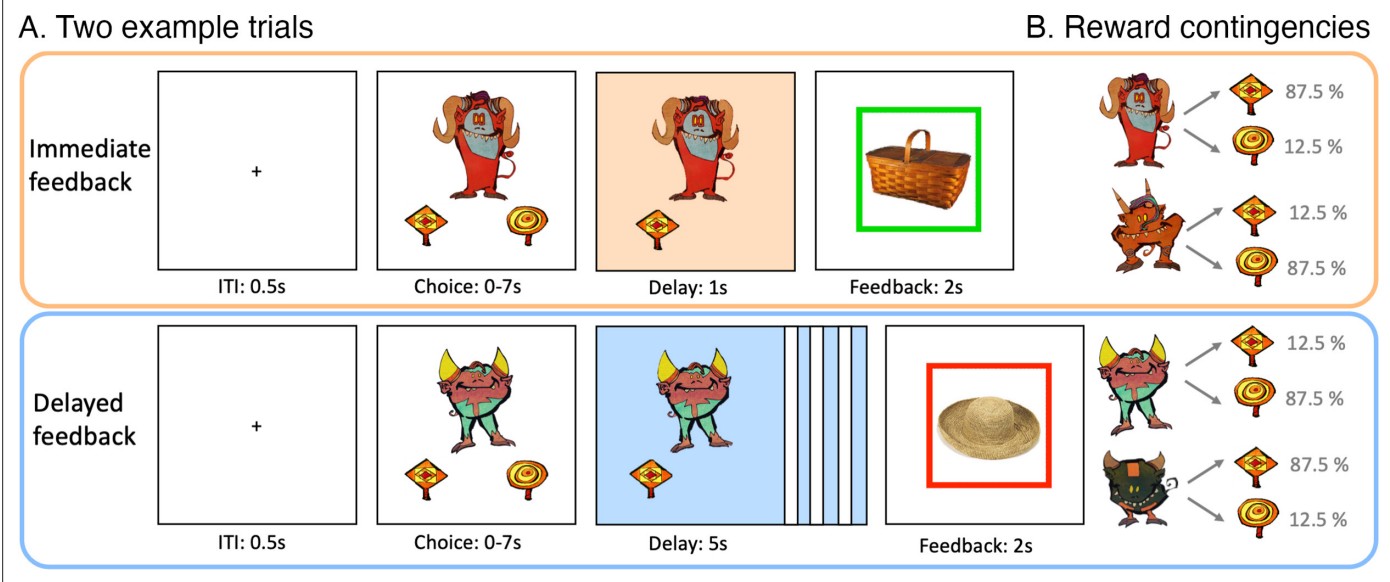

**Figure 1.** Reinforcement learning task. (**A**) Depiction of two example trials of immediate and delayed feedback conditions presented at wave 1. For immediate feedback (top panel), between choice response and feedback, cue and choice were presented for 1 s. At feedback, a green frame around the incidentally encoded object indicated a positive outcome, which appeared in 87.5% of the trials when selecting the squard-shaped lolli for this example cue. For delayed feedback (bottom panel), the delay phase between choice response and feedback lasted for 5 s. The red frame around the object indicated a negative outcome and appeared in 87.5% of the trials when selecting the squard-shaped lolli for this example cue. (**B**) For each feedback condition, two action-outcome contingencies were learned to balance a potential choice bias. With the four task versions, the cues and outcome contingencies were counterbalanced across participants.

showed a more protracted inverted-U-shaped trajectory that peaked in adolescence. Based on these structural findings, striatal and hippocampal systems are expected to develop functionally at different rates (**Lavenex and Banta Lavenex, 2013**), with habit memory depending on the earlier developing striatum and episodic memory depending on the later developing hippocampus (**Mandolesi et al., 2009**). A direct investigation of the longitudinal development of both memory systems in childhood would shed light on whether the memory systems show a differential engagement similar to that of adults (**Foerde and Shohamy, 2011**). Such knowledge could be useful to structure learning processes according to the developmental status. For example, children's ability to learn from delayed feedback may depend on how well their hippocampus has developed. In the same study sample, we previously reported that children's hippocampal volume was related to their family's income level (**Raffington et al., 2019**). Additionally, previous research has shown that stress can reduce the effectiveness of the hippocampal-dependent memory system (**Schwabe and Wolf, 2013**). This suggests that environmental factors such as income and stress may play a role in shaping how well children learn from delayed feedback, particularly through their impact on hippocampal development. By identifying the specific environmental factors that impact children's learning and brain development, we can identify risk groups and tailor interventions to ameliorate adverse effects.

This study aimed to explore the development of value-based learning in children and its relationship with structural brain development over time. We hypothesized that the timing of feedback would modulate children's learning in a commonly used reinforcement learning task (see **Figure 1**), and that such modulation can be captured by reinforcement learning (RL) model parameters. Additionally, we predicted that children's value-based longitudinal development would shift towards more optimal learning behavior. Regarding structural brain development, we expected the striatum to be relatively mature by middle childhood compared to the protracted hippocampal maturation. Our second objective was to investigate the relationship between value-based learning and structural brain development using longitudinal structural equation modeling. We anticipated that there would be differentiated brain-cognition links between brain volume and value-based learning. Specifically, we predicted that immediate feedback learning would be more strongly associated with striatal volume, whereas hippocampal volume would be more closely linked to delayed feedback and the facilitation of episodic memory encoding. Finally, we examined how these brain-cognition dynamics would change over time by analyzing their longitudinal changes.

## Results

### Behavioral results

First, we were interested in whether children showed behavioral differences between waves and feedback timing. A descriptive overview is provided in **Table 1** and **Figure 2**. The details of the reported GLMM models, including the random effects structure and the effects of age and sex, are described in the Appendix 2. Since some children were poor learners who failed to reach 50% average accuracy in their last 20 trials (13 children at wave 1 and 6 children at wave 2), we also performed behavioral analyses with a reduced dataset in which results remained unchanged (Appendix 6).

**Table 1.** Behavioral learning outcomes and mixed model fixed effects that predicted the outcomes.

| | Descriptive Results | | | | Mixed Model Effects | |
|---|---|---|---|---|---|---|
| | Wave 1 | | Wave 2 | | Wave | Feedback |
| | Immediate | Delayed | Immediate | Delayed | | |
| ACC | 0.69 (0.46) | 0.70 (0.46) | 0.79 (0.41) | 0.80 (0.40) | ↑ W2 | – |
| WS | 0.81 (0.39) | 0.80 (0.40) | 0.88 (0.32) | 0.88 (0.32) | ↑ W2 | – |
| LS | 0.47 (0.50) | 0.50 (0.50) | 0.42 (0.49) | 0.42 (0.49) | ↓ W2 | – |
| RT | 2.10 (1.31) | 2.07 (1.29) | 1.70 (1.02) | 1.67 (1.00) | ↓ W2 | ↓ Delayed |

Note. Mean (standard deviation of accuracy) (ACC, probability correct), win-stay probability (WS), lose-shift probability (LS), and reaction time (RT, in seconds), split by wave and feedback timing. Mixed model effects and their directionality of effect (increasing ↑ or decreasing ↓). W2 = Wave 2.

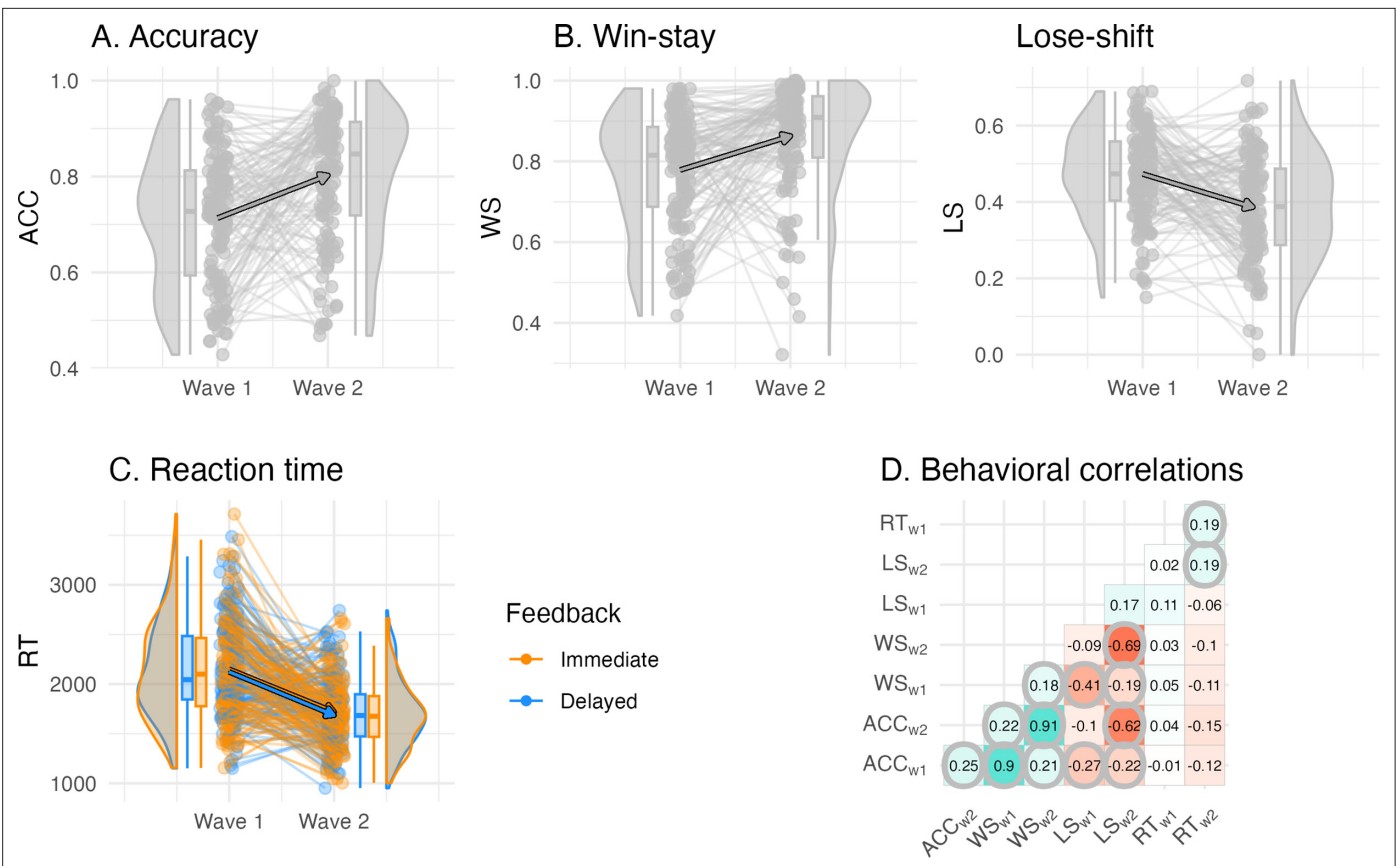

**Figure 2.** Individual differences in the behavioral learning outcomes and their longitudinal change. (**A**) Accuracy did not differ by feedback timing and increased between waves. (**B**) Win-stay and lose-shift proportion did not differ by feedback timing, and win-stay increased and lose-shift proportion decreased between waves. (**C**) Reaction time (in ms) differed by feedback timing, in which decisions for cues learned with delayed feedback were faster, and reaction times were faster at wave 2 compared to wave 1. (**D**) Correlations between behavioral outcomes reveal that learning accuracy was primarily correlated with the win-stay and lose-shift probabilities both within and between waves, but was uncorrelated to reaction time. Significant correlations are circled, p-values were adjusted for multiple comparisons using bonferroni correction.

### Children's learning improved between waves

With the complete dataset, we found that increased learning accuracy (i.e. the probability of choosing the more rewarding option) was predicted at wave 2 compared to wave 1, but there were no differences in accuracy by feedback timing ($\beta_{wave=2}$ = .550, SE = .061, z = 8.97, p < 0.001, $\beta_{feedback=delayed}$ = 0.013, SE = 0.024, z = 0.54, p = 0.590). Furthermore, win-stay probability increased and lose-shift probability decreased longitudinally, again without differences by feedback timing (WS: $\beta_{wave=2}$ = 0.586, SE = 0.071, z = 8.22, p < 0.001, $\beta_{feedback=delayed}$ = 0.023, SE = 0.033, z = 0.69, p = 0.489; LS: $\beta_{wave=2}$ = –0.252, SE = 0.037, z = –6.87, p < 0.001, $\beta_{feedback=delayed}$ = 0.030, SE = 0.022, z = 1.37, p = 0.169). Reaction times were faster at wave 2 compared to wave 1, and they were faster for delayed compared to immediate feedback trials ($\beta_{wave=2}$ = –0.221, SE = 22.8, t(135) = –9.70, p < 0.001, $\beta_{feedback=delayed}$ = –13.8, SE = 6.59, t(136) = –2.10, p = 0.038). To summarize, children's average accuracy improved over 2 years, while their win-stay probability increased and their lose-shift probability decreased between waves. Children were able to respond faster to cues paired with delayed feedback compared to cues paired with immediate feedback, and they became faster in their decision-making across waves (see mixed model effects overview in *Table 1*). Of note, reaction times were largely uncorrelated with accuracy and switching behavior (win-stay, lose-shift), while accuracy and switching behavior showed significant correlations at both waves (*Figure 2D*).

**Table 2.** Model comparison results.

| Model | Parameters | $\Delta elpd_{loo}$ [SE] | $\Sigma elpd_{loo}$ [mean] | Pseudo-BMA+ |
|---|---|---|---|---|
| Step 1: heuristic strategy models and value-based learning model | | | | |
| $vbm_1$ | $1\alpha, 1\tau$ | 0 [0] | −15154.9 [-0.45] | 1 |
| $ws$ | $1\tau_{ws}$ | −1327.7 [159.5] | −16482.7 [-0.49] | < 0.01 |
| $wsls$ | $1\tau_{wsls}$ | −4247.3 [284.8] | −19402.3 [-0.58] | 0 |
| Step 2: value-based learning models | | | | |
| **$vbm_3$** | $1\alpha, 2\tau$ | 0 [0] | −15045.3 [-0.45] | 0.73 |
| $vbm_7$ | $1\alpha, 2\rho$ | −2.93 [2.92] | −15048.2 [-0.45] | 0.24 |
| $vbm_6$ | $2\alpha, 1\rho$ | −24.34 [8.85] | −15069.6 [-0.45] | < 0.01 |
| $vbm_8$ | $2\alpha, 2\rho$ | −29.71 [15.95] | −15075.0 [-0.45] | 0.02 |
| $vbm_4$ | $2\alpha, 2\tau$ | −43.34 [14.89] | −15088.6 [-0.45] | < 0.01 |
| $vbm_2$ | $2\alpha, 1\tau$ | −46.45 [13.97] | −15091.7 [-0.45] | < 0.01 |
| $vbm_5$ | $1\alpha, 1\rho$ | −59.01 [7.59] | −15104.3 [-0.45] | < 0.01 |
| $vbm_1$ | $1\alpha, 1\tau$ | −109.63 [11.98] | −15154.9 [-0.45] | < 0.01 |

Note. Model = heuristic (*ws*, *wsls*) and value-based models (*$vbm_{1-8}$*) that were compared against each other. Parameters = corresponding model parameters learning rate $\alpha$, inverse temperature $\tau$ and outcome sensitivity $\rho$. $\Delta elpd_{loo}$ [SE] = difference in the Bayesian leave-one-out cross-validation estimate of the expected log pointwise predictive density relative to the winning model and its standard errors. $\Sigma elpd_{loo}$ [mean] = sum of expected log pointwise predictive density of all 33,460 trials, including all participants and waves, and trial mean. *Pseudo-BMA+* = model weight for relative model evidence using Bayesian model averaging stabilized by Bayesian bootstrap with 100,000 iterations.

## Modeling results

### Children's behavior was best described by value-based learning

We conducted a 2-step sequential procedure for model development and model selection. Model comparison using leave-one-out cross validation showed evidence in favor of the value-based learning model, reflected in the highest expected log pointwise predictive density and highest model weights, confirming that children's learning behavior in the longitudinal data can generally be better described by a value-based rather than by a heuristic strategy model ($elpd_{loo}$ = −15154.9, *Pseudo-BMA+* = 1, *Table 2*). Children whose individual fit was better for a heuristic model (*wsls*) than for the value-based model (*$vbm_1$*), were at both waves more likely to be poor learners (defined as an accuracy below 50% in the last 20 trials). Taken together, children's learning behavior was best described by a value-based model, and a heuristic strategy model captured more poor learners compared to a value-based model.

**Table 3.** Description of computational model parameters from the winning value-based model $vbm_3$ .

| | Wave 1 | | | | | Wave 2 | | | | |
|---|---|---|---|---|---|---|---|---|---|---|
| | $\alpha$ | $\tau_{Immediate}$ | $\tau_{Delayed}$ | $ls_{Immediate}$ | $ls_{Delayed}$ | $\alpha$ | $\tau_{Immediate}$ | $\tau_{Delayed}$ | $ls_{Immediate}$ | $ls_{Delayed}$ |
| Mean | 0.02 | 14.6 | 14.8 | 0.73 | 0.73 | 0.05 | 16.2 | 16.5 | 0.82 | 0.82 |
| SD | 0.02 | 2.04 | 2.37 | 0.12 | 0.13 | 0.04 | 2.37 | 2.21 | 0.13 | 0.13 |
| Min | < 0.01 | 6.73 | 5.25 | 0.53 | 0.53 | < 0.01 | 4.37 | 6.85 | 0.53 | 0.53 |
| Max | 0.09 | 17.5 | 17.9 | 0.94 | 0.94 | 0.22 | 18.6 | 18.7 | 0.96 | 0.96 |

Note. $\alpha$ = *learning rate across feedback timing*, $\tau/ls$ = *inverse temperature and learning score separated by conditions of feedback timing.*

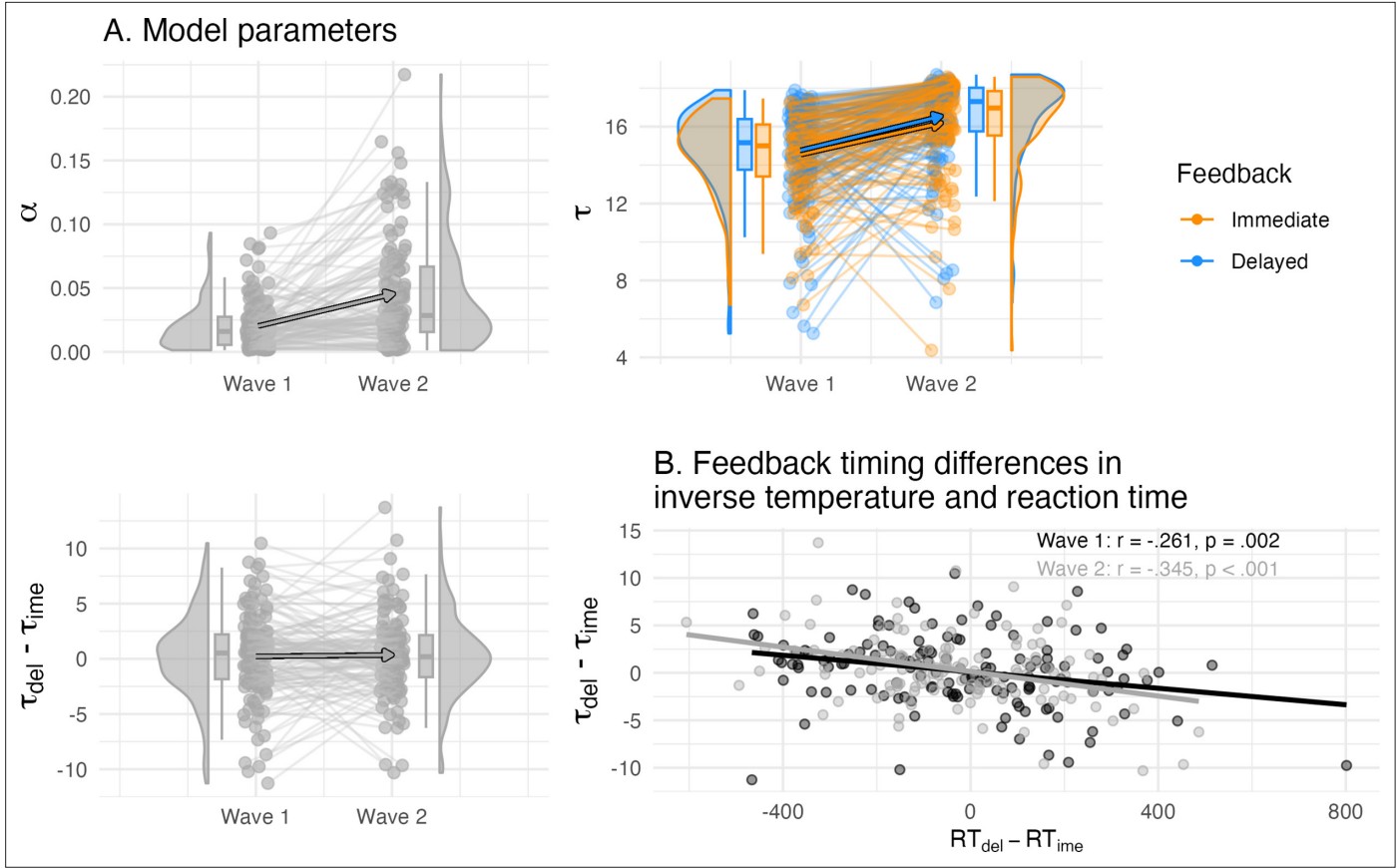

**Figure 3.** Overview of the computational model parameters. (**A**) Individual differences in the learning rate and inverse temperature of the winning model and their longitudinal change. The inverse temperature $\tau$ but not learning rate $\alpha$ was separated by feedback timing, and both increased between waves in their values (top panel). The condition difference in the inverse temperature did not differ on average, but showed individual differences (bottom left panel). (**B**) The condition differences in the inverse temperature correlated with reaction time, that is higher delayed compared to immediate inverse temperature was related to faster delayed compared to immediate reaction time.

## Feedback timing modulated choice stochasticity

Model $vbm_3$ ($1\alpha2\tau$) showed the largest model evidence, reflected in the highest expected log pointwise predictive density and highest model weights and suggests that feedback timing affected the inverse temperature, but not the learning rate or outcome sensitivity ($elpd_{loo} = -15045.3$, *pseudo-BMA+ = 0.73, Table 2*). *Table 3* and *Figure 3A* provide a descriptive overview of the winning model parameters. Of note, there were only small differences in model fit ($elpd_{loo}$) to the second-best model ($vbm_7$, $1\alpha2\rho$, $\Delta elpd_{loo} = -2.93$, $elpd\_SE_{loo} = 2.92$, *Pseudo-BMA+ = 0.24*), which suggests a potential separable feedback timing effect on outcome sensitivity. We also performed the model comparison with a reduced dataset in which the winning model remained the same (Appendix 6). The average inverse temperature did not differ by feedback condition, but showed large within-person condition differences at both waves, indicating individual differences in feedback timing modulation (wave 1: $\Delta\tau_{del-ime}$ Mean = 0.22, SD = 3.80, Range = 21.74, wave 2: $\Delta\tau_{del-ime}$ Mean = 0.35, SD = 3.70, Range = 24.03). The correlations between the parameters are reported in Appendix 3.

Since reaction times were predicted by feedback timing behaviorally, and inverse temperature is assumed to reflect decision-making, we were interested in whether differences in reaction time were related to inverse temperature differences. Indeed, at both waves, children who responded faster during delayed compared to immediate feedback had a higher inverse temperature at delayed compared to immediate feedback (wave 1: $r = -0.261$, $t(138) = -3.18$, $p = 0.002$, wave 2: $r = -0.345$, $t(124) = -4.10$, $p < 0.001$, *Figure 3B*). Taken together, children's learning behavior was best described by a value-based model, where feedback timing modulated individual differences in the choice rule during value-based learning. Interestingly, the differences in the choice rule and reaction time were

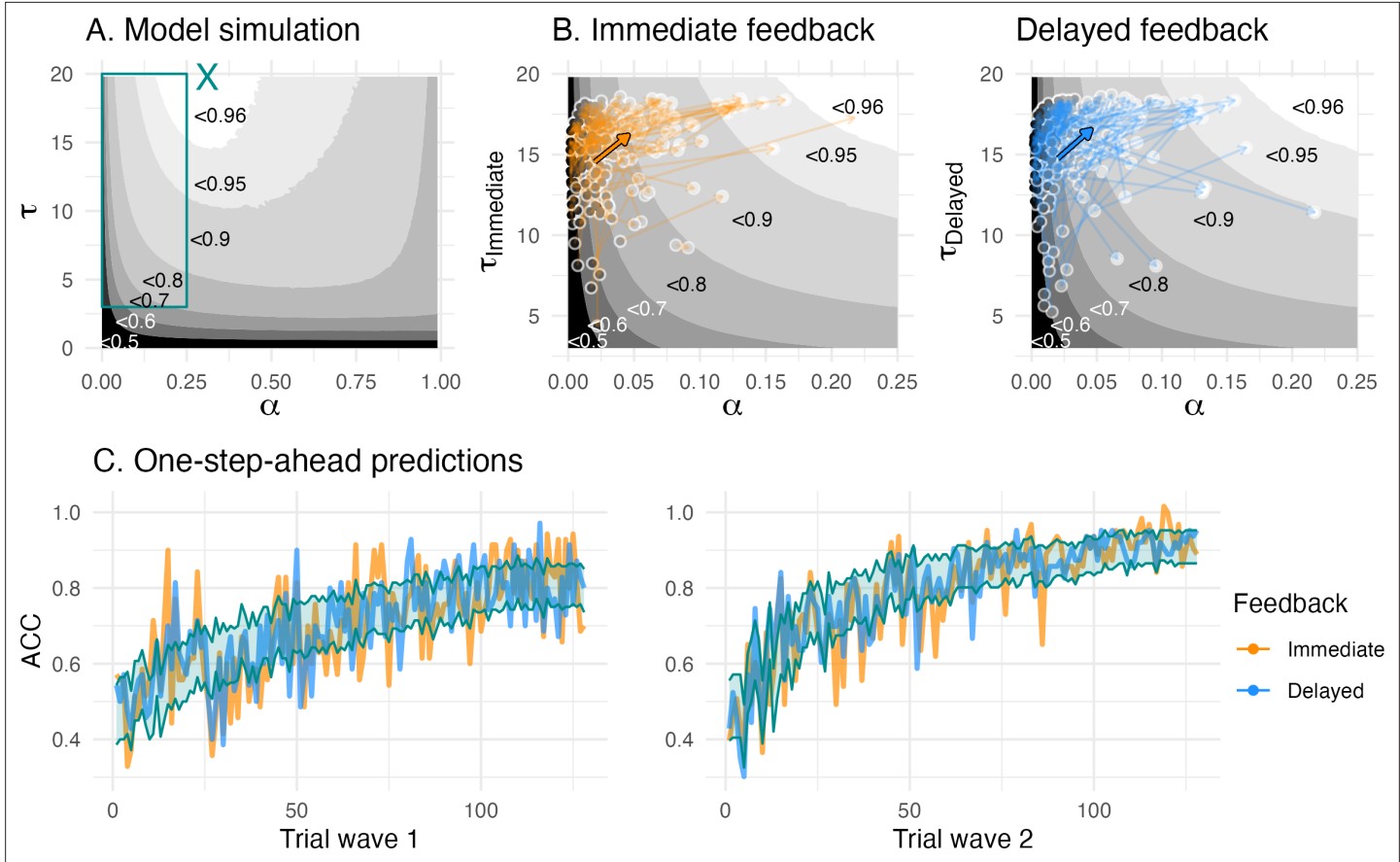

**Figure 4.** Model simulation/validation. (**A**) The model simulation depicts parameter combinations and simulation-based average learning scores. The cyan 'X' in the middle top depicts the optimal parameter combination where average learning scores were at 96.5%, and the cyan rectangle depicts the space of the fitted parameter combinations, (**B**) Enlarged view of the space of fitted parameter combinations. The colored arrows depict mean change (bold arrow) and individual change (transparent arrows) of the fitted parameters. The greyscale gradient-filled dots, that are connected by the arrows, depict the individual learning score, while the the greyscale gradient in the background depicts the simulated average learning score. The mean change reveals an overall change towards the higher, that is, more optimal, learning scores. (**C**) One-step-ahead posterior predictions of the winning model for each wave. The colored lines depict averaged trial-by-trial task behavior for each feedback condition, and a cyan ribbon indicates the 95% highest density interval of the one-step-ahead prediction using the entire posterior distribution, which included 6000 iterations for each of the 33,460 trials.

correlated. Specifically, more value-guided choice behavior (i.e. higher inverse temperature) was related to faster responses during delayed feedback relative to immediate feedback, suggesting a link between model parameter and behavior in relation to feedback timing.

## Children's value-based learning became more optimal

Next, we compared the parameter space according to model simulation (*Figure 4A*) with the empirical posterior parameters fitted by the winning model (*Table 3*, *Figure 4B*) to determine whether children increased their value-based learning towards more optimal parameter combinations. Both fitted and simulated parameter combinations allowed us to derive a learning score that captured learning performance according to the winning value-based model. Note that the learning score was defined as the average choice probability for the more rewarded choice option. We refer to these model-derived choice probabilities as learning score, since they reflect value-based learning and combine information of learned values, that depend on the learning rate, and values translated into choice probabilities, that depend on the inverse temperature. Thus, a higher learning score reflects more optimal value-based learning. We simulated 10,000 parameter combinations and created a learning score map according to each parameter combination (*Figure 4A*). The optimal parameter combination was at a learning rate $\alpha = 0.29$, and an inverse temperature $\tau = 19.8$, and with an average learning score of 96.5% (*Figure 4A*). Children's fitted learning rates ranged 0.01–0.22 and

inverse temperature 6.73–18.70 and were outside the parameter space of a learning score above 96% (*Table 3* and *Figure 4A*). The average longitudinal increases in learning rate and inverse temperature were mirrored by average increases in the learning scores, confirming our prediction that their parameters developed towards optimal value-based learning (arrow in *Figure 4B*). We further found that the average longitudinal change in win-stay and lose-shift proportion also developed towards more optimal value-based learning (Appendix 4).

## Model validation

To validate our winning model $vbm_3$, we estimated its predictive accuracy by comparing one-step-ahead model predictions with the choice data. The one-step ahead predictions of the winning model captured children's choices overall well, with predictive accuracies of 65.3% at wave 1 and 75.7% at wave 2 (*Figure 4C*). Further, our winning model showed a good parameter recovery for learning rate ($r = 0.85$) and inverse temperature ($r = 0.75$–0.77). Our winning model showed excellent on the group level (100%) when comparing it to a set of models used during model comparison ($vbm_1$, $vbm_7$, $wsls$). The individual model recovery was lower (58%), with 35% of the simulated winning model fitting best on our baseline model $vbm_1$ with a single inverse temperature, which likely reflects the noisy property of the inverse temperature (Appendix 1).

## Longitudinal brain-cognition links

### Significant longitudinal change in brain and cognition

We first performed univariate LCS model analyses to estimate a latent change score of immediate and delayed learning scores as well as striatal and hippocampal volumes (see descriptive changes in *Figure 5B, C*). All four variables of interest showed significant positive mean changes and variances, and all univariate models provided a good fit to the data (see Appendix 5). This allowed us to further relate the differences in structural brain changes to changes in learning.

### Hippocampal volume exhibited more protracted development during middle childhood

We next fitted a bivariate LCS model to compare striatal and hippocampal change scores. We theorized that by middle childhood, the striatum would be relatively mature, whereas the hippocampus continues to develop. We progressively constructed multiple LCS models to test this idea. First, the bivariate LCS model provided a good data fit ($\chi^2$ (14) = 10.09, CFI = 1.00, RMSEA (CI) = 0 (0-0.06), SRMR = 0.04). We then further fitted two constrained models, to see whether setting the mean striatal change or the mean hippocampal change to 0 would lead to a drop in the model fit. Compared to the unrestricted model, the constrained model that assumed no striatal change did not lead to a drop in model fit ($\Delta\chi^2$ (1) = 2.74, $p$ = 0.098), whereas the model that assumed hippocampal change dropped in model fit ($\Delta\chi^2$ (1) = 12.69, $p$ < 0.001). Finally, we tested a more stringent assumption of equal change for striatal and hippocampal volumes, in which the model dropped in model fit compared to the unrestricted model ($\Delta\chi^2$ (1) = 18.04, $p$ < 0.001) and suggests that striatal and hippocampal change differed. Together, these results support our postulation of separable maturational brain trajectories in our study sample, suggesting that the hippocampus continued to grow in middle childhood, whereas striatal volume increased less.

### Hippocampal and striatal volume showed distinct associations to learning

We fitted a four-variate LCS model to test our prediction of selective brain-cognition links. Specifically, we assumed a larger contribution of striatal volume at immediate learning, and a larger contribution of hippocampal volume at delayed learning. The LCS model provided good data fit ($\chi^2$ (27) = 15.4, CFI = 1.00, RMSEA (CI) = 0 (0 –0.010, SRMR = 0.045)), and all relevant paths are shown in *Figure 5D* (see *Table 4* for a detailed model overview). For the striatal associations to cognition, we found that wave 1 striatal volume covaried with both immediate learning score and delayed learning score ($\phi_{STR_{w1},LS_{i,w1}}$ = 0.19, $z$ = 2.52, SE = 0.07, $p$ = 0.012, $\phi_{STR_{w1},LS_{d,w1}}$ = 0.18, $z$ = 2.37, SE = 0.07, $p$ = 0.018). Constraining the striatal association to immediate learning to 0 worsened the model fit relative to the unrestricted model ($\Delta\chi^2$ (1) = 5.66, $p$ = 0.017), which was the same when constraining the striatal association to delayed learning to 0 ($\Delta\chi^2$ (df 1) = 5.14, $p$ = 0.023). In summary, larger striatal volume was associated

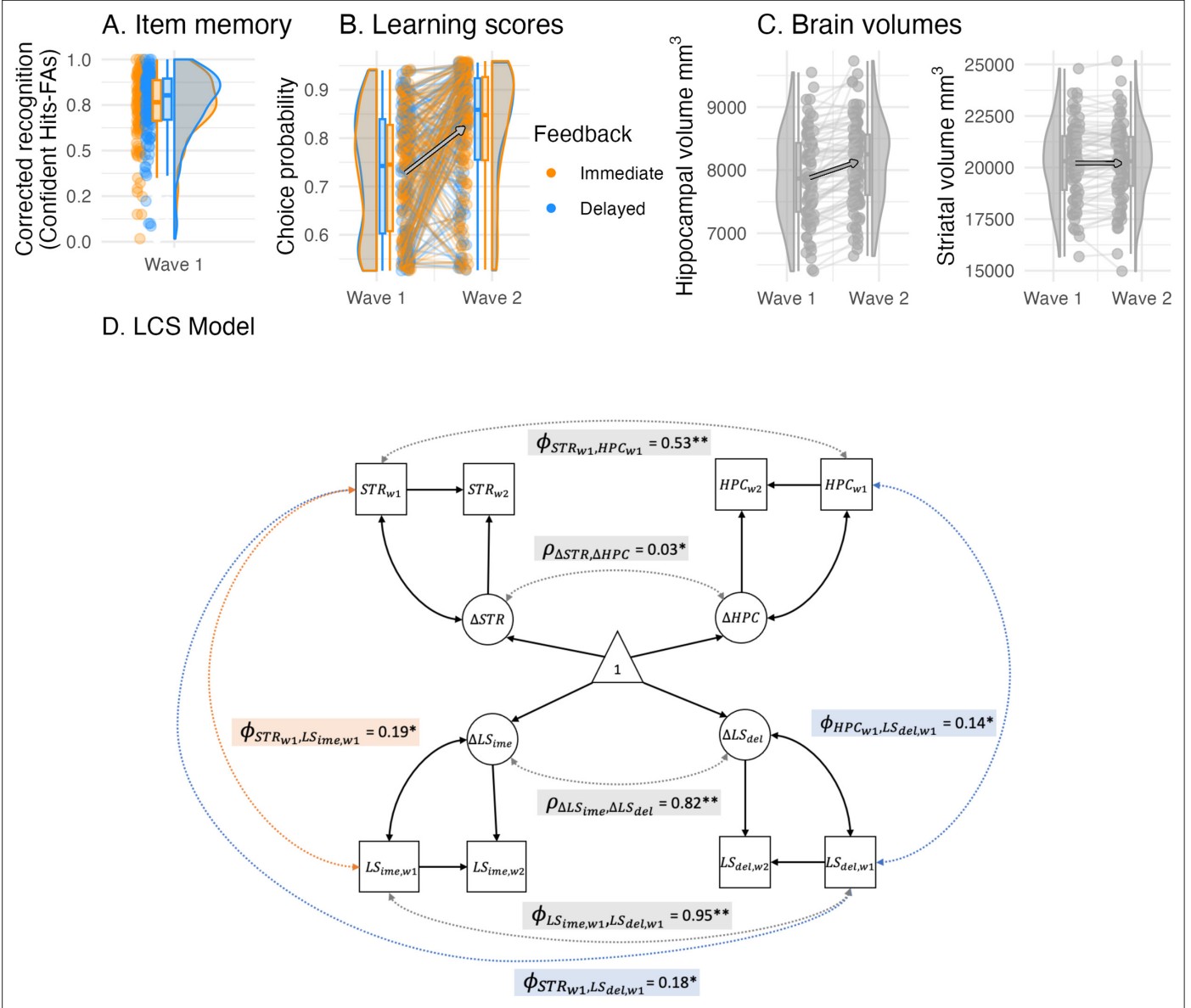

**Figure 5.** Cognitive and brain measures with cross-sectional and longitudinal links. (**A**) Recognition memory (corrected recognition = hits - false alarms) for objects presented during delayed feedback was only enhanced at trend. (**B**) Learning scores depicted here were used in the LCS analyses. Learning scores were the model-derived choice probability of the contingent choice using fitted posterior parameters. (**C**) Hippocampal and striatal volumes increased between waves, while hippocampal volume increased most. (**D**) A four-variate latent change score (LCS) model that included striatal and hippocampal volumes as well as immediate and delayed learning scores. Depicted are significant paths cross-domain (brain-cognition, dashed lines) and within-domain (brain or cognition, solid lines), other paths are omitted for visual clarity and are summarized in *Table 4*. Depicted brain-cognition links included $\phi_{STR_{w1},LS_{ime,w1}}$ (covariance between striatal volume and immediate learning score at wave 1), as well as $\phi_{HPC_{w1},LS_{del,w1}}$ and $\phi_{STR_{w1},LS_{del,w1}}$ (covariances between hippocampal and striatal volumes and delayed learning score at wave 1). Brain links included $\phi_{STR_{w1},HPC_{w1}}$ and $\rho_{\Delta STR,\Delta HPC}$ (wave 1 covariance and change-change covariance), and similarly, cognition links included $\phi_{LS_{ime,w1},LS_{del,w1}}$ and $\rho_{\Delta LS_{ime},\Delta LS_{del}}$. Covariates included age, sex and estimated total intracranial volume. ** denotes significance at $\alpha < 0.001$, * at $\alpha < 0.05$.

with better learning scores for both immediate and better delayed feedback. This pattern remained the same in the results of the reduced dataset (Appendix 6).

Hippocampal volume, on the other hand, only covaried with delayed learning at wave 1 ($\phi_{HPC_{w1},LS_{d,w1}}$ = 0.14, $z$ = 2.05, $SE$ = 0.07, $p$ = 0.041), not with immediate learning score ($\phi_{HPC_{w1},LS_{i,w1}}$ = 0.12, $z$ = 1.68, $SE$ = 0.07, $p$ = 0.092). Fixing the path between hippocampal volume and delayed learning to 0 worsened the model fit relative to the unrestricted model ($\Delta\chi^2$ (1) = 4.19, $p$ = 0.041), but not when its path to immediate learning was constrained to 0 ($\Delta\chi^2$ (1) = 2.94, $p$ = 0.086). This suggests that larger

**Table 4.** Parameter estimates of a four-variate latent change score model that includes brain (striatal and hippocampal volume) and cognition domains (immediate and delayed learning score).

| | STR | $LS_{ime}$ | HPC | $LS_{del}$ |
|---|---|---|---|---|
| Model fit: $\chi^2$ = 15.4, df = 27, CFI = 1, RMSEA (CI) = 0 (0–0.01), SRMR = 0.045 | | | | |
| Mean change Δ | 0.06* (0.03) | 0.76** (0.08) | 0.38** (0.04) | 0.75** (0.08) |
| wave 1 variance σ | fixed to 1 | fixed to 1 | fixed to 1 | fixed to 1 |
| change variance $\sigma_\Delta$ | 0.07** (0.01) | 0.88** (0.10) | 0.18* (0.07) | 0.83** (0.10) |
| Intercept-change regression β | –0.04 (0.04) | –0.83* (0.29) | –0.16* (0.06) | –0.73* (0.27) |
| **Wave 1 covariates** | | | | |
| age onto Intercept $\phi$ | 0.19 (0.10) | –0.05 (0.08) | 0.29* (0.08) | 0.08 (0.08) |
| sex onto Intercept $\phi$ | –0.42** (0.07) | –0.14 (0.07) | –0.47** (0.07) | –0.11 (0.07) |
| eTIV onto Intercept $\phi$ | 0.68** (0.05) | – | 0.70** (0.05) | – |
| **Brain-cognition links (cross-domain)** | $STR - LS_{ime}$ | $STR - LS_{del}$ | $HPC - LS_{ime}$ | $HPC - LS_{del}$ |
| wave 1 covariation $\phi$ | 0.19* (0.07) | 0.18* (0.07) | 0.12 (0.07) | 0.14* (0.07) |
| change-change covariance $\rho$ | < 0.01 (0.03) | < 0.01 (0.03) | –0.06 (0.05) | –0.07 (0.05) |
| wave 1 brain onto cognition change $\gamma$ | 0.25 (0.13) | 0.22 (0.12) | 0.05 (0.11) | 0.06 (0.10) |
| wave 1 cognition onto brain change $\gamma$ | –0.19 (0.13) | 0.21 (0.13) | 0.05 (0.10) | < 0.01 (0.10) |
| **Brain links (within-domain)** | $STR - HPC$ | | | |
| wave 1 covariation $\phi$ | 0.53** (0.07) | | | |
| change-change covariance $\rho$ | 0.03* (0.01) | | | |
| wave 1 striatum onto hippocampal change $\gamma$ | 0.06 (0.05) | | | |
| wave 1 hippocampus onto striatal change $\gamma$ | 0.02 (0.03) | | | |
| **Cognition links (within-domain)** | $LS_{ime} - LS_{del}$ | | | |
| wave 1 covariation $\phi$ | 0.95** (0.10) | | | |
| change-change covariance $\rho$ | 0.82** (0.10) | | | |
| wave 1 $LS_{ime}$ onto $LS_{del}$ change $\gamma$ | –0.07 (0.27) | | | |
| wave 1 $LS_{del}$ onto $LS_{ime}$ change $\gamma$ | 0.06 (0.28) | | | |

Parameter estimates in bold are the paths of interest depicted in **Figure 5D**. Standard errors are shown in parentheses. eTIV = estimated total intracranial volume. ** denotes significance at $\alpha$ < 0.001, * at $\alpha$ < 0.05. sex coded as 1 = girls, –1 = boys.

hippocampal volume was specifically associated with better delayed learning. In the results of the reduced dataset, the hippocampal association to the delayed learning score was no longer significant, suggesting a weakened pattern when excluding poor learners (Appendix 6). It is likely that the exclusion reduced the group variance for hippocampal volume and delayed learning score in the model.

As a next step, the associations between striatum and hippocampus to immediate or delayed learning was directly compared against each other. A model equal-constraining striatal and hippocampal paths to immediate learning ($\Delta\chi^2$ (1) = 0.41, $p$ = 0.521) and another model equal-constraining these paths to delayed learning ($\Delta\chi^2$ (1) = 0.14, $p$ = 0.707) did not lead to a worse model fit compared to the unrestricted model, which suggests that the brain-cognition links have considerable overlap. This is in line with the high wave 1 covariance and change-change covariance within the brain and cognition domain (see *Table 4*). We found no longitudinal links between the brain and cognition domains, which suggests that the found brain-cognition links at wave 1 remained longitudinally stable (see Appendix 5 for an exploratory LCS model that related the model parameters to striatal and hippocampal volume).

Taken together, the confirmatory LCS model results were in line with our predictions of a relatively larger involvement of the hippocampus during delayed feedback learning, but the findings on striatal volume disconfirmed a selective association with immediate feedback learning and suggest a more general role of the striatum in both learning conditions.

## No evidence for enhanced episodic memory during delayed feedback

Finally, we investigated whether a hippocampal contribution at delayed feedback would selectively enhance episodic memory. Episodic memory, as measured by individual corrected object recognition memory (hits - false alarms) of confident ('sure') ratings, showed at trend better memory for items shown in the delayed feedback condition ($\beta_{feedback=delayed}$ = 0.009, *SE* = 0.005, *t*(137) = 1.80, *p* = 0.074, see *Figure 5A*). Note that in the reduced dataset, delayed feedback predicted enhanced item memory significantly (Appendix 6). The inclusion of poor learners in the complete dataset may have weakend this effect because their hippocampal function was worse and was not involved in learning (nor encoding), regardless of feedback timing. To summarize, there was inconclusive support for enhanced episodic memory during delayed compared to immediate feedback, calling for future study to test the postulation of a selective association between hippocampal volume and delayed feedback learning.

## Discussion

In this study, we examined the longitudinal development of value-based learning in middle childhood and its associations with striatal and hippocampal volumes that were predicted to differ by feedback timing. Children improved their learning in the 2-year study period. Behaviorally, learning was improved by an increase in accuracy and a reduction in reaction time (i.e. faster responses). Further, children's switching behavior improved by an increase in win-stay and a decrease in lose-shift behavior. Computationally, learning was enhanced by an increase in learning rate and inverse temperature, which together constituted more optimal value-based learning. Further, feedback timing modulated specifically the inverse temperature. In terms of brain structures, we found that longitudinal changes in hippocampal volume were larger compared to striatal volume, which suggests more protracted hippocampal maturation. The brain-cognition links were longitudinally stable and partially confirmed our hypotheses. In line with previous adult literature and our assumption, hippocampal volume was more strongly associated with delayed feedback learning. Contrary to our expectations, episodic memory performance was not enhanced under delayed feedback compared to immediate feedback. Furthermore, striatal volume unexpectedly was associated with both immediate and delayed feedback learning, suggesting a common involvement of the striatum during value-based learning in middle childhood across timescales.

Children's learning improvement between waves was described behaviorally by increased win-stay and decreased lose-shift behavior. Our finding is in line with cross-sectional studies in the developmental literature that reported increased learning accuracy and win-stay behavior (*Chierchia et al., 2023*; *Habicht et al., 2022*). Our longitudinal dataset with younger children further suggests that learning change is not only accompanied by increased win-stay, but also decreased lose-shift behavior. We found lower learning performance and less optimal switching behavior in girls compared to boys, which could point to sex differences for reinforcement learning during middle childhood (Appendix 2). Previous studies have found both male and female advantages depending on their age and the type of learning task (*Mandolesi et al., 2009*; *Overman, 2004*; *Evans and Hampson, 2015*). Alternatively, sex differences may have been driven by confounding variables not included in the analysis.

Computationally, we found longitudinally increased and more optimal learning rate and inverse temperature, as shown by simulation data, that add to the growing literature of developmental reinforcement learning (*Nussenbaum and Hartley, 2019*). Adult studies that examined feedback timing during reinforcement learning reported average learning rates range from 0.12 to 0.34 (*Foerde and Shohamy, 2011*; *Höltje and Mecklinger, 2020*; *Lighthall et al., 2018*), which are much closer to the simulated optimal learning rates of 0.29 than children's average learning rates of 0.02 and 0.05 at wave 1 and 2 in our study. Therefore, it is likely that individuals approach adult-like optimal learning rates later during adolescence. However, the differences in learning rate across studies have to be interpreted with caution. The differences in the task and the analysis approach may limit their comparability

(*Zhang et al., 2020*; *Eckstein et al., 2021*). Task proporties such as the trial number per condition differed across studies. Our study included 32 trials per cue in each condition, while in adult studies, the trials per condition ranged from 28 to 100 (*Foerde and Shohamy, 2011*; *Höltje and Mecklinger, 2020*; *Lighthall et al., 2018*). Optimal learning rates in a stable learning environment were at around 0.25 for 10–30 trials (*Zhang et al., 2020*), another study reported a lower optimal learning rate of around 0.08 for 120 trials (*Behrens et al., 2007*). This may partly explain why in our case of 32 trials per condition and cue, optimal learning rates called for a relatively high optimal learning rate of 0.29, while in other studies, optimal learning rates may be lower. Regarding differences in the analysis approach, the hierarchical bayesian estimation approach used in our study produces more reliable results in comparison to maximum likelihood estimation (*Brown et al., 2020*), which had been used in some of the previous adult studies and may have led to biased results towards extreme values. Taken together, our study underscores the importance of using longitudinal data to examine developmental change as well as the importance of simulation-based optimal parameters to interpret the direction of developmental change.

Despite a relatively immature hippocampal structure in middle childhood, our results confirmed a longitudinally stable association between hippocampal volume and delayed feedback learning. However, episodic memory in this learning condition was not enhanced. This suggests a developmentally early hippocampal contribution to value-based learning during delayed feedback, which does not modulate episodic memory as much as compared to adults. Therefore, our study partially extends the findings from the adult literature to middle childhood (*Foerde and Shohamy, 2011*; *Foerde et al., 2013*; *Höltje and Mecklinger, 2020*; *Lighthall et al., 2018*). The reduced effect of delayed feedback on episodic memory may be due to the protracted development of hippocampal maturation. In an aging study with a similar task, older adults failed to exhibit enhanced episodic memory for objects presented during delayed feedback trials, and they showed no enhanced hippocampal activation during delayed feedback and (*Lighthall et al., 2018*). Therefore, the findings converge nicely at both childhood and older adulthood, during which the structural and functional integrity of hippocampus are known to be less optimal than at younger adulthood (*Shing et al., 2010*; *Keresztes et al., 2017*; *Ghetti and Bunge, 2012*).

Our brain-cognition links were only partially confirmed, as striatal volumes exhibited associations with not just immediate learning scores, as we predicted, but also with delayed learning scores. This result suggests that the striatum may be important for value-based learning in general rather than exhibiting a selective association with immediate feedback learning. This is also what we found in an explorative analysis that related the striatum to learning rate in general and further predicted longitudinal change in learning rate (Appendix 5). This overall reduced brain-behavior specificity could reflect less differentiated memory systems during development, similar to findings from aging research. Here, older adults exhibited stronger striatal and hippocampal co-activation during both implicit and explicit learning, compared to more dissociable brain-behavior relationships in younger adults (*Dennis and Cabeza, 2011*). Interestingly, even in young adults, clear dissociations between memory systems such as in non-human lesion studies are uncommon, and factors like stress modulate their cooperative interaction (*Packard and Goodman, 2013*; *Packard et al., 2018*; *Schwabe and Wolf, 2013*; *Ferbinteanu, 2016*; *White and McDonald, 2002*). Further, there are methodological differences to previous studies that could explain why striatal volumes were not uniquely associated with immediate learning in our study. For example, previous studies related reward prediction errors to striatal and hippocampal activation (*Foerde and Shohamy, 2011*; *Höltje and Mecklinger, 2020*; *Lighthall et al., 2018*), whereas we examined individual differences in brain structure and the model-derived learning scores. Future functional neuroimaging studies with children could further clarify whether children's memory systems are indeed less differentiated and explain the attenuated modulation by feedback timing. Taken together, compared to the adult literature, our results with children showed that the hippocampal structure was associated with delayed feedback learning, but did not enhance episodic memory encoding, while the striatum generally supported value-based learning. These findings point towards a developmental effect of less differentiated and more cooperative memory systems in middle childhood.

Our computational modeling results revealed a separable effect of feedback timing on inverse temperature, which suggests that the memory systems modulated learning during decision-making. The reported behavioral differences in reaction time and their correlation to the inverse temperature

further support the idea of a decision-related mechanism, as we found children to respond faster during delayed feedback trials and faster responding children also exhibited more value-guided choice behavior (i.e. higher inverse temperature) during delayed compared to immediate feedback. The hippocampus may contribute to a decision-related effect in the delayed feedback condition by facilitating the encoding and retrieval of learned values (*Shadlen and Shohamy, 2016*). This is in contrast to previous event-related fMRI and EEG studies reporting feedback timing modulations at value update (*Foerde and Shohamy, 2011*; *Höltje and Mecklinger, 2020*; *Lighthall et al., 2018*), which may be due to at least two reasons. First, we did not include a functional brain measure to examine its differential engagement during the choice and feedback phases. Second, in such a reinforcement learning task, disentangling model parameters from the choice and feedback phases can be challenging, such as for the inverse temperature and outcome sensitivity (*Browning et al., 2023*). Taken together, hippocampal engagement at delayed feedback may enhance outcome sensitivity as well as facilitate choice behavior through improved retrieval of action-outcome associations. A mechanism facilitating retrieval seems especially relevant in our paradigm, where multiple cues were learned and presented in a mixed order, thus creating a high memory load. To summarize, our study results suggest that feedback timing could modulate decision-making in addition to or as alternative to a mechanism at value update. However, disentangling the effects of inverse temperature and outcome sensitivity is challenging and warrants careful interpretation. Future studies might shed new light by examining neural activations at both task phases, by additionally modeling reaction times using a drift-diffusion approach, or by choosing a task design that allows independent manipulations of these phases and associated model parameters, for example, by using different reward magnitudes during reinforcement learning, or by studying outcome sensitivity without decision-making.

One aim of developmental investigations is to identify the emergence of brain and cognition dynamics, such as the hippocampal-dependent and striatal-dependent memory systems, which have been shown to engage during reinforcement learning depending on the delay in feedback delivery. Our longitudinal study partially confirmed these brain-cognition links in middle childhood but with less specificity as previously found in adults.

An early existing memory system dynamic, similar to that of adults, is relevant for applying reinforcement learning principles at different timescales. In scenarios such as in the classroom, a teacher may comment on a child's behavior immediately after the action or some moments later, in par with our experimental manipulation of 1 s versus 5 s. Within such short range of delay in teachers' feedback, children's learning ability during the first years of schooling may function equally well and depend on the striatal-dependent memory system. However, we anticipate that the reliance on the hippocampus will become even more pronounced when feedback is further delayed for longer time. Children's capacity for learning over longer timescales relies on the hippocampal-dependent memory system, which is still under development. This knowledge could help to better structure learning according to their development. Furthermore, probabilistic learning from delayed feedback may be a potential diagnostic tool to examine the hippocampal-dependent memory system during learning in children at risk. Environmental factors such as stress (*Schwabe and Wolf, 2013*) and socioeconomic status (*Raffington et al., 2019*; *Hackman et al., 2010*) have been shown to affect hippocampal structure and function and may contribute to a heightened risk for psychopathology in the long term (*Frodl et al., 2010*; *Lucassen et al., 2017*; *Rahman et al., 2016*). Deficits in hippocampal-dependent learning may be particularly relevant to psychopathology since dysfunctional behavior may arise from a tendency to prioritize short-term consequences over long-term ones (*Levin et al., 2018*; *Von Siebenthal et al., 2017*) and from the maladaptive application of previously learned behavior in inappropriate contexts (*Maren et al., 2013*). Interestingly, poor learners showed relatively less value-based learning in favor of stronger simple heuristic strategies, and excluding them modulated the hippocampal-dependent associations to learning and memory in our results. More studies are needed to further clarify the relationship between hippocampus and psychopathology during cognitive and brain development.

Another key question is whether developmental trajectories observed cross-sectionally are also confirmed by longitudinal results, such as for the learning rate and inverse temperature. Our results show developmental improvements in these learning parameters in only 2 years. This suggests that the initial 2 years of schooling constitute a dynamic period for feedback-based learning, in which contingent feedback is important in shaping behavior and development.

## Materials and methods

### Participants

Children and their parents took part in 2 waves of data collection with an interval of about 2 years (*Mean* = 2.07, *SD* = 0.17, *Range* = 1.69–2.68). The inclusion criteria for wave 1 were children attending first or second grade, no psychiatric or physical health disorders, at least one parent speaking fluent German, and born full-term (≥37 weeks of gestation). At wave 1, 142 children (46% female, age *Mean* = 7.19, *SD* = 0.46, *R*ange = 6.07–7.98) and their parents or caregivers participated in the study. 140 children were included in the analysis (one child did not complete the probabilistic learning task, and another child was later excluded due to technical problems during the task). A subgroup of 90 children (49% female, 100% right-handed), who was randomly selected, completed magnetic resonance imaging (MRI) scanning at wave 1, and 82 of them contributed to structural data after removing scans with excessive movement. At wave 2, 127 children (46% female, age *Mean* = 9.25, *SD* = 0.45, *Range* = 8.30–10.2) continued taking part in the study, while families of the remaining children were unable to be contacted or decided not to return to the study. A total of 126 children at wave 2 completed the reinforcement learning task and were included in the analysis. All children at wave 2 were invited for MRI scanning, and 104 of them completed scanning (45% female, 92% right-handed). Ninety-nine children contributed to structural data, after removing scans with excessive movement. In total, 73 children contributed to the longitudinal MRI data and 126 children contributed to the longitudinal learning data. As previously reported for this study sample, we found no systematic bias due to wave 2 dropout (*Raffington et al., 2019*).

### Procedure

The study consisted of a series of cognitive tasks tested during two behavioral sessions, including a reinforcement learning task, and one MRI session at wave 1 (*Raffington et al., 2019*; *Raffington et al., 2020*). Two years later, the children underwent one behavioral and one MRI session. MRI scanning was performed within 3 weeks of the behavioral task session. Each session lasted between 150 and 180 min and was scheduled either on weekdays between 2 p.m. and 6 p.m. or during weekends. Before participation, the parents provided written informed consent and children's verbal assent at both waves. All children were compensated with an honorarium of 8 euro per hour.

### Measures

#### Reinforcement learning task

Children completed an adapted reinforcement learning task (*Foerde and Shohamy, 2011*) in which they learned the preferred associations between four cues (cartoon characters) and two choices (round-shaped or square-shaped lolli) through probabilistic feedback (87.5% contingent and 12.5% non-contingent reward probability). In each trial, after an initial inter-trial interval of 0.5 s, a cue and its choice options were presented for up to 7 s until the child made a choice (*Figure 1*, choice phase). In the delay phase, we manipulated feedback timing. For two cues, the selected choice remained visible for 1 s (immediate feedback condition), whereas for the other two cue characters, it remained visible for 5 s before feedback was given (delayed feedback condition). A final feedback phase of 2 s indicated a reward by a green frame, and a punishment by a red frame. Inside each frame, a unique object picture was shown, which was incidentally encoded and irrelevant to the task. The child was instructed to pay attention to the feedback indicated by the frame color. In an initial practice phase of 32 trials, the child practiced the task with a fifth cartoon character not included in the actual task to avoid practice effects. The experimenter instructed the child to select the choice that was most likely to result in a reward. The Experimenter checked whether the child learned the more rewarded choice during practice and let it repeat the practice task otherwise to ensure understanding of the task. In the actual task, 128 trials were presented in four blocks and with small breaks in between. Cues were presented in a mixed, pseudo-randomized order. A total of 64 unique objects were shown in the feedback phase, each one twice within the same feedback condition. In both delay phases, contingent choice and choice location remained the same for each cue within the task, but were balanced across participants by using four different task versions. At wave 2, four new cues replaced the previous ones to rule out memory effects.

## Object recognition test

At wave 1, children were additionally tested for recognition memory on the object pictures that were incidentally encoded during reinforcement learning. A total of 80 objects (48 old objects and 32 new objects) were presented in randomized order. The 48 old objects (24 for each feedback condition) were selected from the 64 old objects shown during learning based on two lists to balance the shown and omitted old objects across task versions. Each old object was shown twice during learning, but if the child failed to respond during learning, no feedback or object was shown in the trial, so some objects only appeared once. These objects were excluded at the individual level (individually missing object *Mean* = 2.71). At recognition, children had 4 response options ('old sure', 'old unsure', 'new unsure', 'new sure') with up to 7 s to respond. The children answered verbally, and the experimenter entered their response. At wave 2, this test was excluded due to time constraints.

## Brain volume

We extracted the bilateral brain volumes for our regions of interest, which were striatum and hippocampus. The striatum regions included nucleus accumbens, caudate and putamen. For our imaging data, structural MRI images were acquired on a Siemens Magnetom TrioTim syngo 3 Tesla scanner with a 12-channel head coil (Siemens Medical AG, Erlangen, Germany) using a 3D T1–weighted Magnetization Prepared Rapid Gradient Echo (MPRAGE) sequence, with the following parameters: 192 slices; field of view = 256 mm, voxel size = 1 mm$^3$, TR = 2500ms; TE = 3.69ms, flip angle = 7°, TI = 1100ms. Volumetric segmentation was performed using the Freesurfer 6.0.0 image analysis suite (*Fischl, 2012*). Previous studies suggested that software tools based on adult brain templates provide inaccurate segmentation for pediatric samples, which can be improved through the use of study-specific template brains (*Phan et al., 2018*; *Schoemaker et al., 2016*). Thus, we created two study-specific template brains (one for each wave) using Freesurfer's 'make_average_subject' command. This pipeline utilized the default adult template brain registrations of the 'recon–all–all' command to average surfaces, curvatures, and volumes from all subjects into a study–specific template brain. All subjects were then re-registered to this study-specific template brain to improve segmentation accuracy. Segmented images were manually inspected for accuracy and 8 cases at wave 1 and 5 cases at wave 2 were excluded for inaccurate or failed registration due to excessive motion.

## Data analysis

### Behavioral learning performance

As a first step, we calculated learning outcomes diretly from the raw data, which where learning accuracy, win-stay and lose-shift behavior as well as reaction time. Learning accuracy was defined as the proportion to choose the more rewarding option, while win-stay and lose-shift refer to the proportion of staying with the previously chosen option after a reward and switching to the alternative choice after receiving a punishment, respectively. We used these outcomes as our dependent variables to examine the effect of the predictors feedback timing (immediate, delayed), wave (1, 2), wave 1 age, and sex (girls, boys), utilizing generalized linear mixed models (GLMM) with the R package lme4 (*Bates et al., 2015*). All reported models included random slopes for within-subject factors feedback timing and wave (see Appendix 2 for the model structure). We systematically tested main effects and interactions between the predictors and their interaction had to statistically improve the predictive ability of the model to be included in the final reported model. All predictor variables were grand-mean-centered to interpret the interaction effects independent from other predictors.

### Reinforcement learning models

As a next step, we used computational modeling to compare the learning models of basic heuristic strategies and value-based learning and to determine the model that could best capture children's trial-by-trial learning behavior. For heuristic strategies, we considered models that reflected a Win-stay-lose-shift (wsls) or a Win-stay (ws) strategy. Win-stay is a heuristic strategy in which the same action is repeated if it leads to a positive outcome in the previous trial, and Win-stay-lose-shift additionally switches to a different action if the previous outcome is negative. Note that these model-based outcomes are not identical to the win-stay and lose-shift behavior that were calculated from the raw data. The use of such model-based measure offers the advantage in discerning the underlying

hidden cognitive process with greater nuance, in contrast to classical approaches that directly use raw behavioral data. The models quantified the learning behavior for each individual $I$ for each cue $c$ and trial $t$. The heuristic models consisted of a weight $w$ that reflected its degree in strategy use. In the case of reward $r = 1$, $w$ was equal to 1 for the chosen option (e.g. choice A), and 0 for the unchosen option (e.g. choice B), thus maximizing win-stay, i.e., choosing A at the subsquent trial $t + 1$:

$$w_{i,c,t+1,A|r=1} = 1 \text{ and } w_{i,c,t+1,B|r=1} = 0 \tag{1}$$

For trials $r = 0$ (applicable only to the wsls model), model weights were the opposite, maximizing lose-shift:

$$w_{i,c,t+1,A|r=0} = 0 \text{ and } w_{i,c,t+1,B|r=0} = 1 \tag{2}$$

The initial weights for both choices were set to $w_{i,c,t=1} = 0.5$. The weight $w$ then scaled the parameter $\tau\_wsls$ or $\tau\_ws$ to estimate the individual strategy use during decision-making. The choice probabilities were calculated using the softmax function, for example., for the chosen option $A$:

$$p(A) = \frac{exp^{w_{i,c,t,A} * \tau\_wsls_i}}{exp^{w_{i,c,t,A} * \tau\_wsls_i} + exp^{w_{i,c,t,B} * \tau\_wsls_i}} \tag{3}$$

Thus, a higher probability of strategy use was reflected by a larger value of $\tau\_wsls$ or $\tau\_ws$.

For value-based learning, we considered a Rescorla-Wagner model and several variants based on our theoretical conceptions. The baseline value-based model $vbm_1$ updated the value $v$ of the selected choice ($A$ or $B$) for the next trial $t$. This value update was determined by calculating the difference between the received reward $r$ and the expected value $v$ of the selected choice, which was the reward prediction error. The value update was further scaled by a learning rate $\alpha$ $(0 < \alpha < 1)$:

$$v_{i,c,t+1,A} = v_{i,c,t,A} + \alpha_i (r_{i,c,t} - v_{i,c,t,A}) \tag{4}$$

When the outcome sensitivity parameter $\rho$ $(0 < \rho < 20)$ was included, the reward was additionally scaled at the value update:

$$v_{i,c,t+1,A} = v_{i,c,t,A} + \alpha_i (\rho_i * r_{i,c,t} - v_{i,c,t,A}) \tag{5}$$

The inverse temperature parameter $\tau$ $(0 < \tau < 20)$ was included in the softmax function to compute choice probabilities:

$$p(A) = \frac{exp^{v_{i,c,t,A} * \tau_i}}{exp^{v_{i,c,t,A} * \tau_i} + exp^{v_{i,c,t,B} * \tau_i}} \tag{6}$$

Note, however, that outcome sensitivity and inverse temperature are difficult to fit simultaneously due to non-identifiability issues (**Brown et al., 2021**). Therefore, models including the inverse temperature fixed outcome sensitivity at 1 (inverse temperature model family), assuming no individual differences in outcome sensitivity. For the outcome sensitivity model family, outcome sensitivity was freely estimated, and the inverse temperature was fixed at 1, asssuming the same degree of value-based decision behavior across individuals. Even though outcome sensitivity is usually restricted to an upper bound of 2 to not inflate outcomes at value update, this configuration led to ceiling effects in outcome sensitivity and non-converging model results. Further, this issue was not resolved when we fixed the inverse temperature at the group mean of 15.47 of the winning inverse temperature family model. It may be that in children, individual differences in outcome sensitivity are more pronounced, leading to more extreme values. Therefore, we decided to extend the upper bound to 20, parallel to the inverse temperature, and all our models converged with Rhat < 1.1. Each model family consisted of 4 model variants $vbm_{1-4}$ $(1\alpha 1\tau, 2\alpha 1\tau, 1\alpha 2\tau, 2\alpha 2\tau)$ and $vbm_{5-8}$ $(1\alpha 1\rho, 2\alpha 1\rho, 1\alpha 2\rho, 2\alpha 2\rho)$, in which each parameter was either separated by feedback timing or kept as a single parameter across feedback conditions. Our baseline value-based model $vbm_1$ included a single learning rate and a single inverse temperature $(1\alpha 1\tau)$.

## Parameter estimation

All choice data were fitted in a hierarchical Bayesian analysis using the Stan language in R (*Stan Development Team, 2021*; *R Development Core Team, 2021*) adopted from the hBayesDM package (*Ahn et al., 2017*). Posterior parameter distributions were estimated using Markov chain Monte Carlo (MCMC) sampling running four chains each with 3000 iterations, using the first half of the chain as warmup, and group-level parameters and individual-level parameters were estimated simoultaneously. The hierarchical Bayesian approach provides more stable and reliable parameter estimates as opposed to point-estimation approaches like maximum likelihood estimation (*Brown et al., 2020*). Each model fit both wave 1 and wave 2 data at once, considering the correlation structure of the same parameter across waves, to account for within-subject dependency using the Cholesky decomposition. The Cholesky decomposition used a Lewandowski-Kurowicka-Joe prior of 2, and all other group-level parameters had a prior normal distribution, Normal (0, 0.5). Non-response trials (wave 1 = 2.41%, wave 2 = 0.97% on average) were excluded in advance.

## Model simulation and model-derived learning score

To appropriately interpret the parameter results with respect to the optimal parameter combination of the winning model, we simulated 5,000,000 individual datasets using 10,000 different parameter value combinations (covering the whole range of each parameter) to identify the optimal parameter combination of the winning model that was selected by model comparison. In addition, we computed the model-derived mean choice probability of the contingent, that is, the more rewarded option, and we referred to it as the model-derived learning score. This model-derived choice probability differs from the observed empirical choice probability (i.e. the accuracy of selecting the more rewarded option), because the model-derived learning score combines the model with the data by incorporating latent information carried out by key learning parameters. Thus, the learning score captures observed behavior based on trial-by-trial latent processes predicted by value-based models. We used this as metric to interpret the fitted posterior parameters in relation to the optimal parameter combination of our probabilistic learning task.

## Model selection and validation

We conducted a two-step sequential procedure for the model development and model selection. As a first step, we compared model evidence for the baseline value-based model that does not separate learning rate and inverse temperature by feedback timing ($vbm_1$:$1\alpha, 1\tau$) to the non-value-based, heuristic strategy models that reflect Win-stay or Win-stay-lose-shift strategy behavior (*ws*, *wsls*). As a second step, we compared model evidence for 8 value-based model variants, 4 of the model family with learning rate and inverse temperature ($1\alpha1\tau, 2\alpha1\tau, 1\alpha2\tau, 2\alpha2\tau$) and 4 of the model family with learning rate and outcome sensitivity ($1\alpha1\rho, 2\alpha1\rho, 1\alpha2\rho, 2\alpha2\rho$). This allowed us to compare whether children showed separable effects of feedback timing on one of the model parameters. We compared the model fit using Bayesian leave-one-out cross-validation and obtained the expected log pointwise predictive density ($elpd_{loo}$) using the R package loo (*Vehtari et al., 2017*). We further computed the model weights (*Pseudo-BMA+*) using Pseudo Bayesian model averaging stabilized by Bayesian bootstrap with 100,000 iterations (*Yao et al., 2018*). To validate our models, we estimated predictive accuracy by comparing one-step-ahead model predictions with the choice data (*Zhang et al., 2020*; *Crawley et al., 2020*). We performed parameter recovery for the winning model and model recovery by comparing it to a set of models used during model comparison (Appendix 1; *Wilson and Collins, 2019*).

## Episodic memory at wave 1

We predicted the individual corrected recognition memory (hits-false alarms) by feedback condition in a linear mixed effects model using the R package lme4 (*Bates et al., 2015*). Only confident ('sure') ratings were included in the analysis, which were 98.1% of all given responses. A total of 140 children completed the recognition memory test and 138 were included in the analysis, with two being excluded due to negative corrected recognition memory value (i.e. poor recognition memory). Age and sex were controlled for as covariates.

## Longitudinal brain-cognition links

We used latent change score (LCS) models to examine the longitudinal relationships between brain and learning score measures. LCS models are longitudinal structural equation models that have been widely applied to estimate developmental changes and coupling effects across domains such as the brain and cognition (*Kievit et al., 2018*; *Ferrer and McArdle, 2010*). LCS models allow the definition of specific paths between multiple variables to test explicit hypotheses and estimate latent change from the observed variables that account for measurement error and increase testing power (*van der Sluis et al., 2010*). We compiled univariate LCS models for each variable separately (learning scores and brain volumes) to examine whether there was significant individual variance and change, which could be related within a multivariate LCS model as a next step. Model fit had to be at least acceptable, with a comparative fit index (*CFI*) >0.95, standardized root mean square residual (*SRMR*) < 0.08 and root mean square error of approximation (*RMSEA*) < 0.08 (*Little, 2013*). Age and sex were included as covariates at wave 1, as well as the estimated total intracranial volume (eTIV) when brain volume was included in the model. Multivariate LCS models allow to estimate meaningful brain-cognition relationships: a wave 1 covariance between brain and cognition, brain predicting change onto cognition, or vice versa, and a covariance in both brain and cognition change scores (wave 1 to wave 2). Before compiling the variables into an LCS model, they were checked for outliers ± 4 *SD* around the mean. We identified one outlier for the learning rate at wave 2, which was removed for the explorative LCS model that included model parameters. There were no further outliers in other cognitive variables or brain volumes. Continuous variables were standardized to the wave 1 measure so that wave 2 values represent the change from wave 1, sex was contrast-coded (girls = 1, boys = –1).

## Acknowledgements

We thank the Max Planck Institute for Human Development and all members of the Jacobs study team for their vital contribution, and all participants and family members for taking part in the study.

This study was supported by the Jacobs Foundation [grant 2014-1151] to LR, YLS and CH. The work of YLS was also supported by the European Union (ERC-2018-StG-PIVOTAL-758898), the Deutsche Forschungsgemeinschaft (German Research Foundation, Project ID 327654276, SFB 1315, 'Mechanisms and Disturbances in Memory Consolidation: From Synapses to Systems'), and the Hessisches Ministerium für Wissenschaft und Kunst (HMWK; project 'The Adaptive Mind').

## Additional information

### Funding

| Funder | Grant reference number | Author |
|---|---|---|
| Jacobs Foundation | 2014-1151 | Laurel Raffington Christine Heim Yee Lee Shing |
| Hessisches Ministerium für Wissenschaft und Kunst | "The Adaptive Mind" | Johannes Falck Jochen Triesch Yee Lee Shing |
| European Research Council | ERC-2018-StG-PIVOTAL-758898 | Yee Lee Shing |
| Deutsche Forschungsgemeinschaft | 327654276 | Yee Lee Shing |
| Deutsche Forschungsgemeinschaft | SFB 1315 "Mechanisms and Disturbances in Memory Consolidation: From Synapses to Systems" | Yee Lee Shing |
| Max Planck Society | | Laurel Raffington Yee Lee Shing |

| Funder | Grant reference number | Author |
|--------|------------------------|--------|
| Austrian Science Fund | FWF-M3166 | Lei Zhang |

The funders had no role in study design, data collection and interpretation, or the decision to submit the work for publication.

### Author contributions
Johannes Falck, Conceptualization, Data curation, Software, Formal analysis, Validation, Investigation, Visualization, Methodology, Writing – original draft, Project administration, Writing – review and editing; Lei Zhang, Software, Formal analysis, Validation, Methodology, Writing – review and editing; Laurel Raffington, Formal analysis, Funding acquisition, Investigation, Project administration, Writing – review and editing; Johannes Julius Mohn, Data curation, Investigation, Writing – review and editing; Jochen Triesch, Methodology, Writing – review and editing; Christine Heim, Conceptualization, Resources, Funding acquisition, Writing – review and editing; Yee Lee Shing, Conceptualization, Resources, Supervision, Funding acquisition, Methodology, Writing – original draft, Project administration, Writing – review and editing

### Author ORCIDs
Johannes Falck ⓘ https://orcid.org/0000-0003-0505-0798
Lei Zhang ⓘ https://orcid.org/0000-0002-9586-595X
Laurel Raffington ⓘ http://orcid.org/0000-0002-0144-5605
Johannes Julius Mohn ⓘ http://orcid.org/0000-0002-3893-8008
Jochen Triesch ⓘ http://orcid.org/0000-0001-8166-2441
Christine Heim ⓘ http://orcid.org/0000-0002-6580-6326
Yee Lee Shing ⓘ http://orcid.org/0000-0001-8922-7292

### Ethics
The study was approved by the 'Deutsche Gesellschaft für Psychologie' ethics committee (YLS_012015). Parents provided written informed consent and consent to publish.

Reviewer #1 (Public Review): https://doi.org/10.7554/eLife.89483.3.sa1
Reviewer #2 (Public Review): https://doi.org/10.7554/eLife.89483.3.sa2
Author response https://doi.org/10.7554/eLife.89483.3.sa3

## Additional files

### Supplementary files
• MDAR checklist

### Data availability
Data and code to reproduce analysis results and figures are available at: https://osf.io/pju65/.

The following dataset was generated:

| Author(s) | Year | Dataset title | Dataset URL | Database and Identifier |
|-----------|------|---------------|-------------|-------------------------|
| Falck J, Shing YL | 2024 | Longitudinal Reinforcement Learning in Middle Childhood | https://doi.org/10.17605/OSF.IO/PJU65 | Open Science Framework, 10.17605/OSF.IO/PJU65 |

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

# Appendix 1

## Parameter and model recovery

We simulated 1000 datasets (50 groups, 20 datasets each) using a wide distribution within the boundaries for learning rate (boundaries = [0,1], *Mean* = 0.5, *SD* = 0.25) and inverse temperature (boundaries = [0,20], *Mean* = 10, *SD* = 5). We first performed a parameter recovery to see how well the winning model recovers the simulated parameters (*Appendix 1—figure 1*). Both inverse temperature and learning rate were recovered overall well, with correlations of 0.75–0.77 for the inverse temperature, their condition differences correlating 0.78–0.79, and the learning rates correlating at 0.85. Inverse temperature values were slightly overestimated until a value of 12 and clearly underestimated above 12. The underestimation was less pronounced for the inverse temperature condition differences. Learning rate was also less biased – here, values below 0.5 slightly overestimates and underestimated with values above 0.5. This means that more extreme values, i.e. those closer to the boundaries, were recovered closer towards the group mean. We next performed model recovery to see how well the model evidence is recovered compared to other models that were used during model comparison. Of all 10 models that were used, we performed model recovery on the two best models (winning model $vbm_3$, $1\alpha$, $2\tau$ and second-best model $vbm_7$, $1\alpha$, $2\rho$), our value-based baseline model ($vbm_1$, $1\alpha$, $1\tau$) and our heuristic strategy model (*Appendix 1—figure 2*). We examined recovery on the group and individual level. On the group level, we used the model weight *Pseudo-BMA* +model for relative model evidence using Bayesian model averaging. On the individual level, we used model fit $elpd_{loo}$, which is the individual summed expected log pointwise predictive density of all trials. On the group level, model recovery was excellent, as all models were recovered with model weights of 0.99–1.00. On the individual level, model recovery was lower for the value-based models, with model weights of 0.58–0.83. Specifically, the models $vbm_1$ and $vbm_3$, which only differed in whether inverse temperature was estimated separate by learning condition (immediate and delayed feedback) or across learning condition, were affected. Here, 35% of the datasets that were simulated using separate inverse temperature fitted best on the model with one inverse temperature (and 30% vice versa), and likely reflects the noisy property of the inverse temperature.

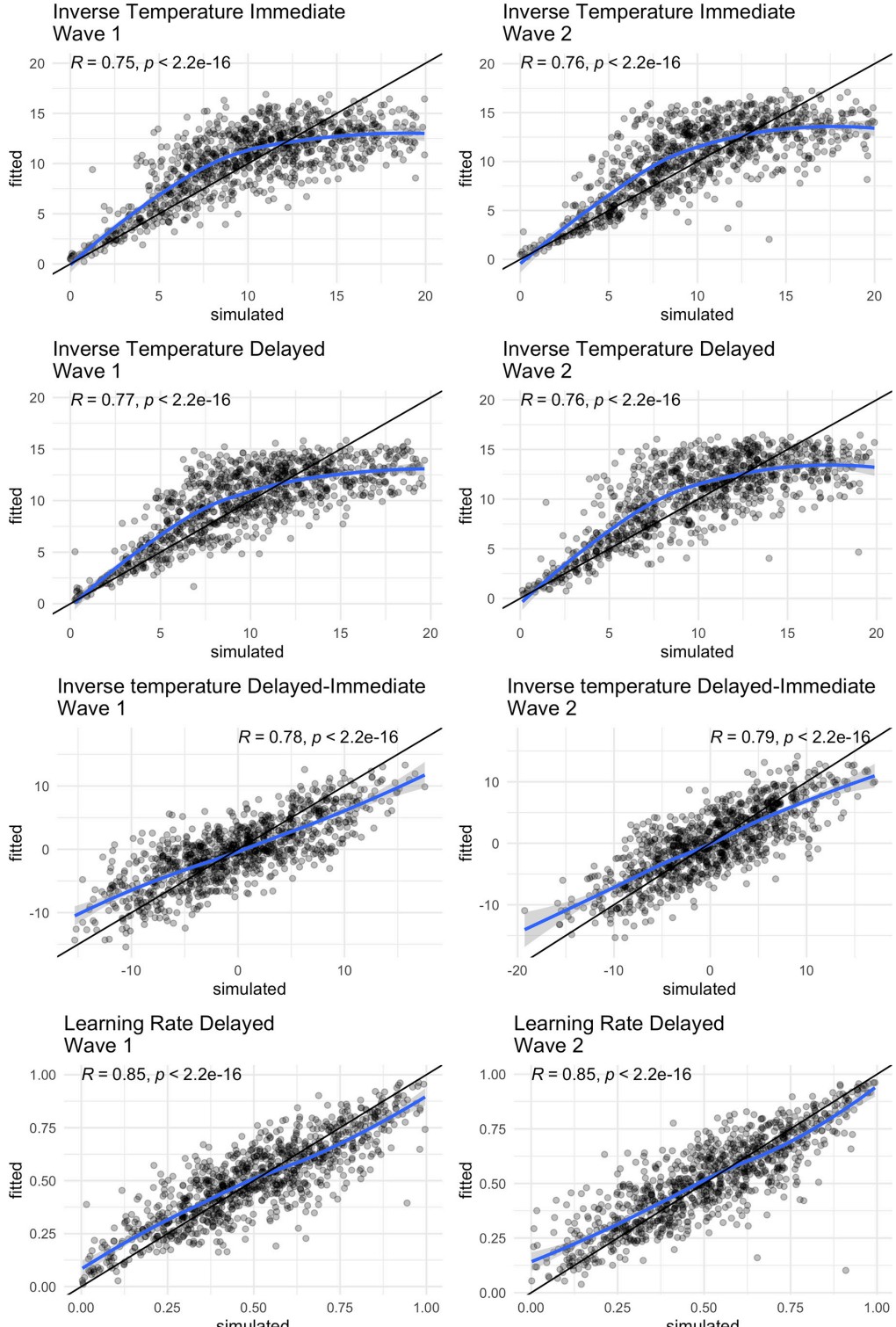

**Appendix 1—figure 1.** Parameter recovery of the winning model, the black line represents the identity line, whereas the blue line is loess regression line, Correlations are calculated by Pearson's r.

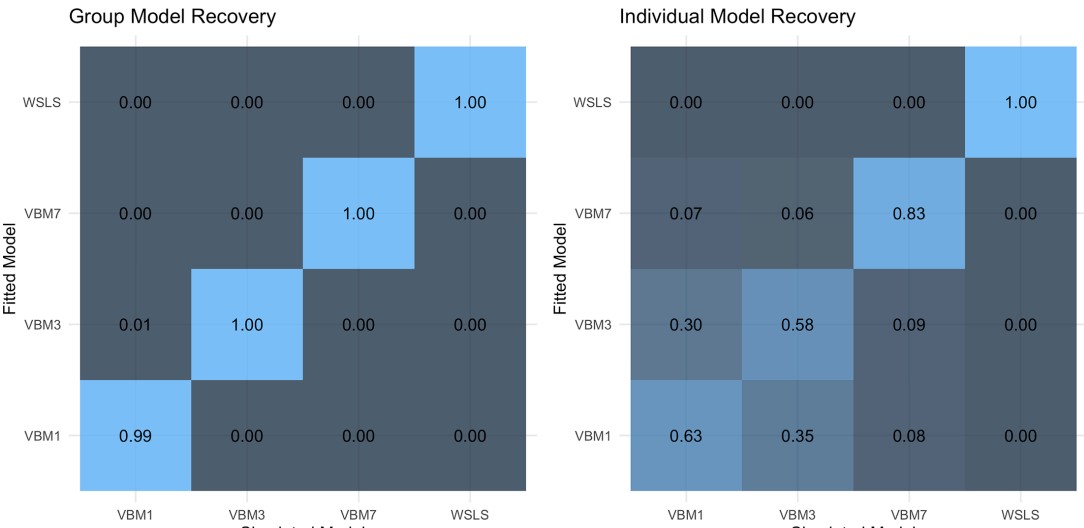

**Appendix 1—figure 2.** Model recovery on the group (left) and individual level (right). Group-level recovery values are the average model weights (across 20 groups, 50 datasets each) *Pseudo-BMA+*using Bayesian model averaging stabilized by Bayesian bootstrap using 100,000 iterations. Individual-level recovery values are the average model fits (across 1000 datasets), $elpd_{loo}$ which is the individual summed expected log pointwise predictive density of all trials.

# Appendix 2

## Model structure and detailed results of generalized linear mixed models (GLMM)

### GLMM Random effects model structure

We ran four GLMMs with the dependent variables accuracy (1 = correct, 0 = incorrect), win-stay behavior (1 = win stay, 0 = win-shift), lose-shift behavior (1 = lose-shift, 0 = lose stay) and reaction time (in milliseconds) as the dependent variable (*Appendix 2—table 1*). As fixed effects, we included within-subject factors wave (1 = wave 1, 2 = wave 2) and feedback type (1 = immediate, 2 = delayed) as well as the covariate sex (1 = girl, 2 = boy). The contrasts of the categorical variables were set using the contr.sum function to keep the mean intercept at the global mean. We first tested whether including the main effects of wave, feedback type and sex improved the model fit. We then tested whether including interaction terms between these three variables, and the model had to improve the overall model fit to be reported as the winning model. As random effects, data were clustered at the participant and learning block level, allowing fixed intercept for each of the 4 blocks (32 trials each) of each individual. As random slopes, we included within-subject factors wave and feedback type.

**Appendix 2—table 1.** Mixed effects model structure and fixed effects results for the models using the dependent variables Accuracy (ACC), win-stay (WS), lose-shift (LS) and Reaction time (RT).

| Fixed effects | $GLMM_{ACC}$ | $GLMM_{WS}$ | $GLMM_{LS}$ | $GLMM_{RT}$ |
|---|---|---|---|---|
| Feedback = Delayed | 0.013 | 0.023 | –0.030 | 14.0* |
| Wave = 2 | 0.550** | 0.586** | –0.252** | –218** |
| Sex = Girls | –0.172* | –0.177* | 0.062 | 23.5 |
| Wave 1 Age | 0.142* | 0.163* | –0.100* | –24.5 |
| Wave = 1*Sex = Girls | not included | not included | 0.068* | not included |
| **Random slopes** | | | | |
| Feedback Type | X | X | X | X |
| Wave | X | X | X | X |
| **Random intercepts** | | | | |
| Participant ID | X | X | X | X |
| Block | X | X | X | X |
| **Model fit** | | | | |
| ICC | 0.44 | 0.45 | 0.12 | 0.23 |
| Observations | 33,460 | 22,013 | 10,383 | 33,460 |
| Marginal $R^2$ | 0.056 | 0.063 | 0.021 | 0.036 |
| Conditional $R^2$ | 0.472 | 0.482 | 0.138 | 0.258 |

Note. ** denotes significance at $\alpha < 0.001$, * at $\alpha < 0.05$. X indicates which random effects were included in the final model. ICC = intraclass correlation. Marginal $R^2$ = variance explained by fixed effects, Conditional $R^2$ = variance explained by fixed and random effects.

### Detailed GLMM results

With the complete dataset, we found that increased learning accuracy was predicted at wave 2 compared to wave 1 ($\beta_{wave=2}$ = 0.550, *SE* = 0.061, z = 8.97, p < 0.001) and with higher age at wave 1 ($\beta_{wave\,1\,age}$ = 0.142, *SE* = 0.070, z = 2.03, p = 0.043), but there were no differences in accuracy by feedback timing ($\beta_{feedback=delayed}$ = 0.013, *SE* = 0.024, z = 0.54, p = 0.590). Girls were overall less accurate than boys ($\beta_{sex=girls}$ = –0.172, *SE* = 0.070, z = 2.45, p = 0.014). Win-stay probability was predicted to be higher at wave 2 ($\beta_{wave=2}$ = 0.586, *SE* = 0.071, z = 8.22, p < 0.001) and with higher age at wave 1 ($\beta_{wave\,1\,age}$ = 0.177, *SE* = 0.078, z = 2.27, p = 0.024), again without differences by feedback timing ($\beta_{feedback=delayed}$ = –0.023, *SE* = 0.032, z = –0.69, p = 0.489). Win-stay probability was lower for girls compared to boys ($\beta_{sex=girls}$ = –0.177, *SE* = 0.078, z = –2.27, p = 0.024). The predicted

Lose-shift probability was lower at wave 2 compared to wave 1 ($\beta_{wave=2}$ = –0.586, $SE$ = 0.071, $z$ = –8.22, p < 0.001) and with higher age at wave 1 ($\beta_{wave\,1\,age}$ = –0.177, $SE$ = 0.078, $z$ = 2.27, $p$ = 0.024), but did not differ by feedback type ($\beta_{feedback=delayed}$ = 0.036, $SE$ = 0.020, $z$ = 1.74, $p$ = 0.081) and sex ($\beta_{sex=girls}$ = 0.063, $SE$ = 0.036, $z$ = 1.76, $p$ = 0.079). Taken together, children on average improved their accuracy, while win-stay probability increased and lose-shift probability decreased between waves. Girls were on average less accurate, showed reduced win-stay behavior and a smaller decrease in lose-shift probability between waves (*Appendix 2—table 1* and *Appendix 2—figure 1*).

Reaction times were predicted to be faster at wave 2 compared to wave 1 ($\beta_{wave=2}$ = –218, $SE$ = 22.7, $t(126)$ = –9.61, $p$ < 0.001), but did not differ by wave 1 age ($\beta_{age\,wave\,1}$ = –42.5, $SE$ = 25.7, $t$ = –1.66, $p$ = 0.100), and they were faster for delayed compared to immediate feedback trials ($\beta_{feedback=delayed}$ = –14.0, $SE$ = 6.61, $t$ = –2.12, $p$ = 0.036). Girls were not different compared to boys ($\beta_{sex=girls}$ = 23.5, $SE$ = 25.7, $t$ = 0.91, $p$ = 0.362). To summarize the reaction time results, children were able to respond faster to cues paired with delayed feedback, compared to cues paired with immediate feedback, and they became faster in their decision making across waves.

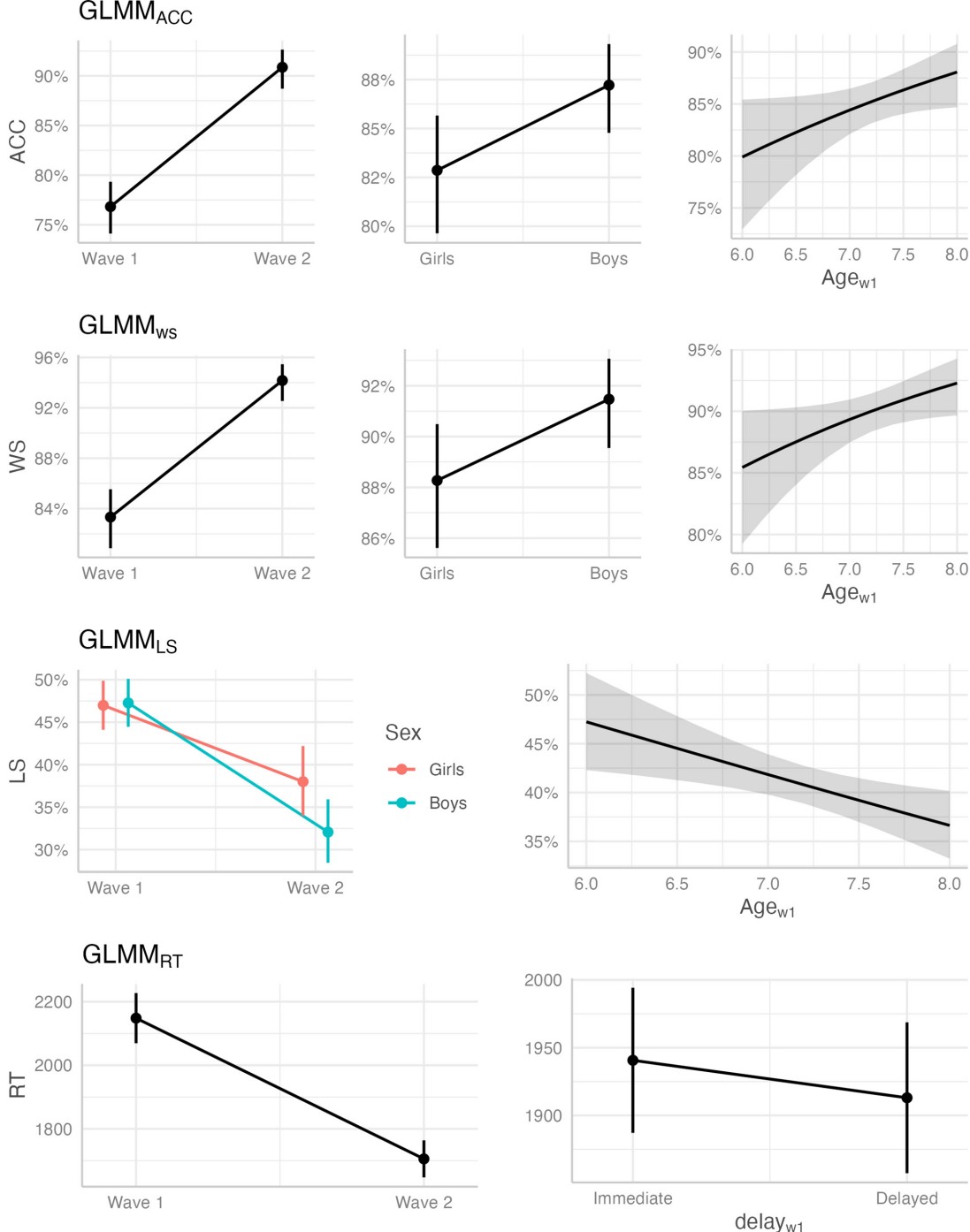

**Appendix 2—figure 1.** Fixed effects plots of significant predictors across behavioral variables accuracy (ACC), win-stay (WS), lose-shift (LS) and reaction time (RT). See *Appendix 2—table 1* for the statistical results.

## Appendix 3

### Winning model parameter correlations

Parameter correlations of the winning model

Correlations between the model parameters learning rate and inverse temperature were only small ($r = 0.19$–$0.25$), which suggests relative independence of these parameters (*Appendix 3—figure 1*). Negative correlations between feedback conditions ($r = -0.31$ to $-0.48$), captured by the inverse temperature, suggest individual differences feedback timing modulation. Positive correlations of the parameters across waves ($r = 0.39$–$0.52$) were moderate to large which suggest temporal stability and showed the appropriateness of our modeling endeavour to incorporate the within-subject data structure. Only inverse temperature for delayed feedback learning was not correlated across waves, which suggests greater temporal instability. Taken together, children's learning behavior was best described by a value-based model, where feedback timing modulated individual differences in the choice rule during value-based learning. Interestingly, differences in the choice rule and reaction times were correlated. Specifically, more value-guided choice behavior (i.e. higher inverse temperature) was related to faster responses during delayed feedback relative to immediate feedback, suggesting a link between model parameter and behavior in relation to feedback timing.

## Parameter correlations

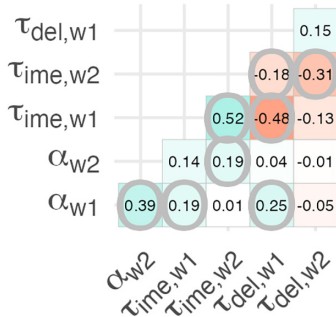

**Appendix 3—figure 1.** Parameter correlations of the winning model. Significant correlations are circled, p-values were adjusted for multiple comparisons using bonferroni correction.

## Appendix 4

### Longitudinal change in Win-Stay and Lose-Shift proportion
### Children's switching behavior became more optimal

In addition to our finding that the change in children's learning rate and inverse temperature became more optimal according to the value-based learning model, we explored whether their change towards optimality is also reflected in children's switching behavior.

We simulated 10,000 parameter combinations and created a learning score map according to each combination of win-stay and lose-shift proportions (*Appendix 4—figure 1*). The optimal proportion for win-stay and lose-shift were at 100% and 24%, respectively. Therefore, both the average longitudinal increase in win-stay proportion (wave 1: 80%, wave 2: 88%) and the average decrease in lose-shift proportion (wave 1: 48%, wave 2: 42%) reflect a change towards more optimal value-based learning.

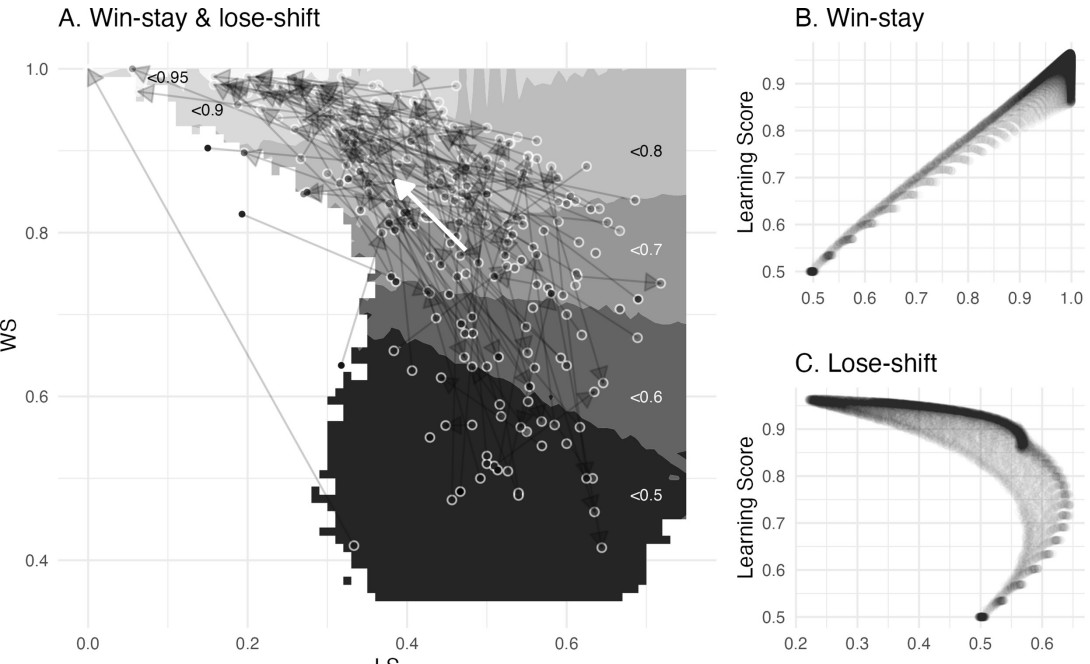

**Appendix 4—figure 1.** Switching behavior and optimal learning derived from model simulation. (**A**) The arrows depict mean change (bold white) and individual change (transparent black) of the empirical win-stay and lose-shift proportions. The greyscale gradient-filled dots, that are connected by the arrows, depict the individual learning score, while the the greyscale gradient in the background depicts the simulated average learning score. The mean change reveals an overall change towards the higher, that is more optimal, learning scores, with higher win-stay and lower lose-shift behavior. (**B, C**) Win-stay and lose-shift behavior plotted against the learning score depict their separate effects on learning optimality. While win-stay showed a positive linear relationship with the learning score, lose-shift showed a negative nonlinear relationship with a larger optimal range.

## Appendix 5

### Confirmatory and exploratory brain-cognition links

This section provides further details on the latent change score (LCS) models from the analysis and provides further LCS models to explore brain-cognition links in the second-best fitting model and to explore the associations with the model parameters learning rate and inverse temperature.

### Univariate LCS models

The model fit and model parameters of the univariate LCS models of our variables of interest (striatal volume, hippocampal volume, immediate learning score, delayed learning score) are summarized in *Appendix 5—table 1*. Of note, learning scores were negatively covaried with sex at wave 1, suggesting reduced immediate learning scores ($\phi_{sex=girls,LS_{i,w1}}$ = –0.20, $z$ = –2.39, $SE$ = 0.08, $p$ = 0.017) and reduced delayed learning scores in girls ($\phi_{sex=girls,LS_{d,w1}}$ = –0.17, $z$ = –2.01, $SE$ = 0.08, $p$ = 0.044).

**Appendix 5—table 1.** Model fit and parameter estimates of the univariate LCS models for immediate and delayed feedback learning score as well as for striatal (STR) and hippocampal (HPC) brain volumes.

|  | $LS_{immediate}$ | $LS_{delayed}$ | **STR** | **HPC** |
|---|---|---|---|---|
| $\chi^2$ (*df*) | 1.75 (4) | 1.25 (4) | 1.61 (6) | 1.77 (6) |
| *RMSEA* (*CI*) | 0.08 (0–0.08) | 0 (0–0.07) | 0 (0–0) | 0 (0–0.02) |
| *SRMR* | 0.03 | 0.03 | 0.03 | 0.03 |
| CFI | 1.00 | 1.00 | 1.00 | 1.00 |
| Mean change $\mu_\Delta$ | 0.74** (0.09) | 0.73** (0.08) | 0.06* (0.03) | 0.37** (0.05) |
| w1 variance $\sigma_\beta$ | 0.99** (0.08) | 0.99** (0.07) | 0.51** (0.07) | 0.46** (0.06) |
| Change variance $\sigma_\Delta$ | 0.94** (0.10) | 0.89** (0.10) | 0.07** (0.02) | 0.18* (0.08) |
| Intercept-change regression $\delta$ | –0.69** (0.08) | –0.73** (0.08) | –0.04 (0.04) | –0.12* (0.04) |
| Age onto Intercept | –0.07 (0.08) | 0.11 (0.08) | 0.02 (0.09) | 0.15 (0.08) |
| Sex onto Intercept | –0.20* (0.08) | –0.17* (0.08) | –0.05 (0.09) | –0.09 (0.09) |
| eTIV onto Intercept | – | – | 0.67** (0.09) | 0.62** (0.10) |

Standard errors in parentheses. ** denotes significance at $\alpha$ < .001, * at $\alpha$ < .05. sex coded as 1 = girls, –1 = boys.

### Confirmatory brain-cognition links with learning scores using the second best fitting model

We fitted a fourvariate LCS model using the second best fitting model to check whether separating outcome sensitivity by feedback timing would show results comparable to those of the winning model that separated inverse temperature by immediate and delayed feedback condition. Using the model-derived learning scores from the second best fitting model, our LCS model again provided a good data fit ($\chi^2$ (27) = 10.1, *CFI* = 1.00, *RMSEA* (*CI*) = 0 (0–0, *SRMR* = 0.042)). However, the brain-cognition links at baseline were not significant for both striatal volume ($\phi_{STR_{w1},LS_{i,w1}}$ = 0.14, $z$ = 1.66, $SE$ = 0.09, $p$ = 0.098 and $\phi_{STR_{w1},LS_{d,w1}}$ = 0.14, $z$ = 1.55, $SE$ = 0.09, $p$ = 0.121) and hippocampal volume ($\phi_{HPC_{w1},LS_{i,w1}}$ = 0.09, $z$ = 1.04, $SE$ = 0.09, $p$ = 0.297 and $\phi_{HPC_{w1},LS_{d,w1}}$ = 0.11, $z$ = 1.22, $SE$ = 0.09, $p$ = 0.222), suggesting no brain-cognition links at wave 1. Longitudinally, striatal volumes predicted larger gains in immediate learning scores ($\beta_{STR_{w1},\Delta ls_i}$ = 0.17, $z$ = 1.97, $SE$ = 0.08, $p$ = 0.049), but this effect diminished when excluding poor learners ($\beta_{STR_{w1},\Delta ls_i}$ = 0.11, $z$ = 1.35, $SE$ = 0.08, $p$ = 0.177). The failure to capture brain-cognition links and the relatively lower model evidence compared to the winning model during model comparison overall suggests that modulations by feedback timing could be captured better by the decision-related parameter inverse temperature rather than by the valuation-related parameter outcome sensitivity.

## Exploratory brain-cognition links with model parameters

The model parameters all showed significant mean change and variance (learning rate: $\mu_{\Delta\alpha}$ = 1.29, $z$ = 7.41, $SE$ = 0.17, p < 0.001, $\sigma_{\Delta\alpha}$ = 3.73, $z$ = 6.77, $SE$ = 0.55, $p$ < 0.001; immediate inverse temperature: $\mu_{\Delta\tau_i}$ = 0.82, $z$ = 9.65, $SE$ = 0.09, p < 0.001, $\sigma_{\Delta\tau_i}$ = 0.97, $z$ = 4.12, $SE$ = 0.24, $p$ < 0.001; delayed inverse temperature: $\mu_{\Delta\tau_d}$ = 0.84, $z$ = 3.91, $SE$ = 0.08, $p$ < 0.001, $\sigma_{\Delta\tau_d}$ = 0.84, $z$ = 3.91, $SE$ = 0.22, $p$ < 0.001). To further understand how the found links between striatal volumes and immediate learning and between hippocampal volumes and delayed learning could be understood as effects of the model parameters, we compiled a five-variate model including brain volumes, learning rates ($\alpha$) and inverse temperature ($\tau$) for immediate and delayed learning. The LCS again provided a good data fit ($\chi^2$ (25) = 15.8, $CFI$ = 1.00, $RMSEA$ ($CI$) = 0 (0 –0.023, $SRMR$ = 0.040)).

For hippocampal volume, we found a positive covariance with delayed inverse temperature at wave 1($\phi_{HC_{w1},\tau_{del,w1}}$ = 0.13, $z$ = 2.30, $SE$ = 0.06, $p$ = 0.021), whereas striatal volume positively covaried with learning rate at ($\phi_{STR_{w1},\alpha_{w1}}$ = 0.15, $z$ = 2.05, $SE$ = 0.08, $p$ = 0.041). The striatal link to learning rate however was diminished when excluding children below the learning criterion. Longitudinally, striatal volume at wave 1 further predicted positive gains in learning rate ($\beta_{STR_{w1},\Delta\alpha}$ = 0.44, $z$ = 2.25, $SE$ = 0.20, $p$ = 0.024). Changes in learning rate covaried positively with changes in immediate inverse temperature ($\phi_{\Delta STR,\Delta\tau_i}$ = 0.35, $z$ = 2.46, $SE$ = 0.14, $p$ = 0.014), while changes in immediate inverse temperature covaried negatively with changes in delayed inverse temperature ($\phi_{\Delta\tau_i,\Delta\tau_d}$ = –0.28, $z$ = –3.60, $SE$ = 0.08, $p$ < 0.001). Immediate inverse temperature at wave 1 predicted negative striatal volume change ($\beta_{\tau_{i,w1},\Delta STR}$ = –0.09, $z$ = –2.38, $SE$ = 0.04, $p$ = 0.017), while delayed inverse temperature at wave 1 predicted negative change in hippocampal volume ($\beta_{\tau_{d,w1},\Delta HPC}$ = –0.08, $z$ = –2.06, $SE$ = 0.04, $p$ = 0.039) in the reduced sample, but not in the full sample. Taken together, while hippocampal volume was only linked to delayed inverse temperature at wave 1, striatal volume was linked to learning rate at wave 1 and was predictive of learning rate development. Further, there was evidence that inverse temperature was predictive of brain volume change in line with the hypothesized brain-cognition links. The inverse temperature between delayed and immediate feedback showed diverging changes, in which the change in immediate inverse temperate was similar to that of learning rate, but dissimilar to that of delayed inverse temperature. This suggests that the hippocampus might be uniquely associated with inverse temperature during delayed learning, whereas the striatum was linked to learning rates, inverse temperature and suggest a stronger contribution to the longitudinal change of learning function in general.

## Appendix 6

### Results when using the reduced dataset

To validate our results, we examined whether the poor learning performance of some of the children in the reinforcement learning task influenced our findings. Therefore, we repeated the analyses with a reduced dataset that excluded children performing below 50% accuracy in their last 20 trials. 13 out of 140 children at wave 1 (54% girls), as well as 6 out of 126 at wave 2 (67% girls) did not reach the learning criterion (above 50% learning accuracy during the last 20 trials of the task) and were excluded in the reduced dataset. In this section, the results are structured into behavioral results, computational modeling results and latent change score modeling results at the end. Whenever there were differences between using the complete and reduced dataset, they were mentioned in the main text and referred to this section for further details.

### Behavioral results

We kept the same model structure to directly compare the results. The fixed effects remained unchanged in all models. All model results remained consistent when using the reduced dataset, with no differences compared to the results obtained using the complete dataset. An overview of the fixed effects and their comparison to the results of the complete dataset are shown in *Appendix 6—table 1*. Using the reduced dataset, the learning accuracy model did not differ in the results, accuracy was predicted by wave ($\beta_{wave=2}$ = 0.492, $SE$ = 0.062, $z$ = 7.88, p < 0.001) and by wave 1 age ($\beta_{age\ wave\ 1}$ = 0.174, $SE$ = 0.071, $z$ = 2.48, $p$ = 0.013), there were no differences by feedback timing ($\beta_{feedback=delayed}$ = 0.009, $SE$ = 0.025, $z$ = 0.35, $p$ = 0.727), and girls were less accurate ($\beta_{sex=girls}$ = –0.157, $SE$ = 0.071, $z$ = –2.18, $p$ = 0.027). The win-stay model also did not differ in the results using the reduced dataset. Win-stay probability was again predicted to be higher at wave 2 ($\beta_{wave=2}$ = 0.534, $SE$ = 0.073, $z$ = 7.27, p < 0.001) and by higher wave 1 age ($\beta_{age\ wave\ 1}$ = 0.186, $SE$ = 0.079, $z$ = 2.36, $p$ = 0.018), there were no differences by feedback timing ($\beta_{feedback=delayed}$ = 0.022, $SE$ = 0.035, $z$ = 0.63, $p$ = 0.531), and girls had a lower win-stay probability ($\beta_{sex=girls}$ = –0.161, $SE$ = 0.080, $z$ = –2.02, $p$ = 0.043). The lose-shift model did not differ using the reduced dataset, lose-shift probability was lower at wave 2 ($\beta_{wave=2}$ = –0.252, $SE$ = 0.037, $z$ =–6.87, p < 0.001), did not differ by feedback type ($\beta_{feedback=delayed}$ = 0.030, $SE$ = 0.022, $z$ = 1.38, $p$ = 0.169) and sex ($\beta_{gender=girls}$ = 0.062, $SE$ = 0.038, $z$ = 1.63, $p$ = 0.102), but the decrease in lose-shift behavior between waves again was smaller for girls ($\beta_{sex=girls\ X\ wave=2}$ = 0.068, $SE$ = 0.034, $z$ = 2.02, $p$ = 0.044). The reaction times were faster at wave 2 compared to wave 1 ($\beta_{wave=2}$ = –221, $SE$ = 23.5, $t$ = –9.42, p < 0.001), they were not predicted by wave 1 age ($\beta_{age\ wave\ 1}$ = –38.0, $SE$ = 26.5, $p$ = 0.154), and they were faster at delayed compared to immediate feedback ($\beta_{feedback=delayed}$ = –16.8, SE = 6.72, $t$ = –2.50, $p$ = 0.014). Girls were not different compared to boys ($\beta_{sex=girls}$ = 20.6, $SE$ = 26.3, $t$ = 0.78, $p$ = 0.436). The magnitude of the fixed effects were overall comparable, only in the accuracy and win-stay model, marginal R² and fixed effects were slightly weaker, which is to be expected when excluding poor learners. To conclude, the behavioral effects remained the same when using the reduced dataset.

**Appendix 6—table 1.** Comparison of the fixed effects results for the models with the reduced and with the complete dataset, each with the dependent variables accuracy (ACC), win-stay (WS), lose-shift (LS) and reaction time (RT).

| Fixed effects | GLMM$_{ACC}$ | GLMM$_{WS}$ | GLMM$_{LS}$ | GLMM$_{RT}$ |
|---|---|---|---|---|
| *Reduced dataset (complete dataset)* | | | | |
| Feedback = Delayed | 0.009 (0.013) | 0.022 (0.023) | –0.030 (–0.030) | –16.8* (–13.8*) |
| Wave = 2 | 0.492** (0.550**) | 0.534** (0.586**) | –0.252** (–0.252**) | –221** (–221**) |
| Sex = Girls | –0.157* (–0.172*) | –0.161* (–0.177*) | 0.062 (0.062) | 20.6 (20.5) |
| Wave 1 Age | 0.174** (0.142*) | 0.186* (0.163*) | –0.100* (–0.100*) | –38.0 (-37.8) |
| Wave = 1*Sex = Girls | not included | not included | 0.068* (0.068*) | not included |
| Model fit | | | | |
| ICC | 0.45 (0.44) | 0.45 (0.45) | 0.12 (0.12) | 0.24 (0.23) |
| Observations | 31857 (33460) | 21212 (22013) | 10383 (10383) | 31857 (33460) |

*Appendix 6—table 1 Continued on next page*

*Appendix 6—table 1 Continued*

| Fixed effects | GLMM$_{ACC}$ | GLMM$_{WS}$ | GLMM$_{LS}$ | GLMM$_{RT}$ |
| --- | --- | --- | --- | --- |
| Marginal R$^2$ | 0.047 (0.056) | 0.054 (0.063) | 0.024 (0.024) | 0.038 (0.036) |
| Conditional R$^2$ | 0.473 (0.472) | 0.483 (0.482) | 0.138 (0.138) | 0.266 (0.260) |

Note. ** denotes significance at α < 0.001, * at α < 0.05. X indicates which random effects were included in the final model. ICC = intraclass correlation. Marginal R$^2$ = variance explained by fixed effects, Conditional R$^2$ = variance explained by both fixed and random effects.

## Model results

We repeated model comparison with the reduced dataset by excluding the $elpd_{loo}$ (expected log pointwise predictive density) of the poor learners (*Appendix 6—table 2*). One may argue that this procedure is suboptimal, as the model parameters were fitted using the complete dataset so that poor learners impacted the parameters of the remaining participants in hierarchical model estimation. However, fitting the reduced dataset only would have required a different model structure, as the amount of longitudinal datasets had been much smaller, and some participants would only have wave 2 data. Since we used a wide prior for model estimation, the impact of poor learners on the group level is reduced.

**Appendix 6—table 2.** Model comparison results obtained with the reduced dataset and the complete dataset.

| Model | Parameters | $\Delta elpd_{loo}$ | mean $elpd_{loo}$ | *Pseudo-BMA+* |
| --- | --- | --- | --- | --- |
| *Reduced dataset (complete dataset)* | | | | |
| *step 1: heuristic strategy vs value-based learning model* | | | | |
| *vbm$_1$* | $1\alpha, 1\tau$ | 0 (0) | –0.47 (-0.45) | 1 (1) |
| *ws* | $1\tau_{ws}$ | –1296.2 (-1327.7) | –0.51 (-0.49) | 0 (< 0.01) |
| *wsls* | $1\tau_{wsls}$ | –4164.3 (-4247.3) | –0.61 (-0.58) | 0 (0) |
| *step 2: value-based learning model variants* | | | | |
| **vbm$_3$** | $1\alpha, 2\tau$ | **0 (0)** | **–0.47 (-0.45)** | **0.78 (0.73)** |
| *vbm$_7$* | $1\alpha, 2\rho$ | –3.71 (-2.93) | –0.47 (-0.45) | 0.19 (0.24) |
| *vbm$_6$* | $2\alpha, 1\rho$ | –24.34 (-24.34) | –0.47 (-0.45) | < 0.01 (< 0.01) |
| *vbm$_8$* | $2\alpha, 2\rho$ | –29.20 (-29.71) | –0.47 (-0.45) | 0.02 (0.02) |
| *vbm$_4$* | $2\alpha, 2\tau$ | –43.86 (-43.34) | –0.47 (-0.45) | < 0.01 (< 0.01) |
| *vbm$_2$* | $2\alpha, 1\tau$ | –45.08 (-46.45) | –0.47 (-0.45) | < 0.01 (< 0.01) |
| *vbm$_5$* | $1\alpha, 1\rho$ | –57.65 (-59.01) | –0.47 (-0.45) | < 0.01 (< 0.01) |
| *vbm$_1$* | $1\alpha, 1\tau$ | –107.8 (-109.63) | –0.47 (-0.45) | < 0.01 (< 0.01) |

Note. Model = Heuristic (**ws**, **wsls**) and value-based models (**vbm$_{1-8}$**) that were compared against each other. Parameters = corresponding model parameters learning rate ($\alpha$), inverse temperature ($\tau$) and outcome sensitivity ($\rho$). $\Delta elpd_{loo}$ = differences in Bayesian leave-one-out cross-validation estimate of the expected log pointwise predictive density relative to the winning model and its standard errors. **mean $elpd_{loo}$** = mean of expected log pointwise predictive density across all trials. *Pseudo-BMA+* = model weight for relative model evidence using Bayesian model averaging stabilized by Bayesian bootstrap using 100,000 iterations.

The model comparison of the reduced dataset did not differ from the results of the complete dataset. At the first step, children's learning behavior in the longitudinal data again can be better described by a value-based rather than by a heuristic strategy model. At the second step, comparison different value-based models, the winning model again suggests that feedback timing affected the inverse temperature, but not the learning rate or outcome sensitivity. We did not find any deviations from the findings of the winning model when using the reduced dataset. The mean model fit (*mean $elpd_{loo}$*) was slightly worse in the reduced dataset, which suggests that the additional poor learners

in the complete dataset did not fit worse to the model than the other children, despite their low accuracy. The correlations between condition differences of inverse temperature and reaction times remained ($r = –0.288$, $t(125) = –3.36$, $p = 0.001$ at wave 1 and $r = –0.352$, $t(118) = –4.09$, $p < 0.001$ at wave 2). To conclude, the same winning model from the computational analysis remained and was therefore used for further analyses.

## Confirmatory brain-cognition links with learning scores and episodic memory

We fitted a fourvariate LCS model using the reduced dataset to check whether the reported results remained the same. The LCS again provided a good data fit ($\chi^2$ (27) = 18.7, *CFI* = 1.00, *RMSEA* (*CI*) = 0 (0 –0.030, *SRMR* = 0.053)). Striatal volume at wave 1 again covaried with both immediate and delayed learning score ($\phi_{STR_{w1},LS_{i,w1}} = 0.17$, $z = 2.19$, *SE* = 0.08, $p = 0.029$ and $\phi_{STR_{w1},LS_{d,w1}} = 0.16$, $z = 2.04$, *SE* = 0.08, $p = 0.041$). Constraining the striatal association to immediate learning to 0 worsened model fit relative to the unrestricted model ($\Delta\chi^2$ (1) = 3.96, $p = 0.047$), but not when constraining the striatal association to delayed learning to 0 ($\Delta\chi^2$ (1) = 3.58, $p = 0.058$). Hippocampal volume did not covary with any learning scores in the reduced dataset ($\phi_{HPC_{w1},LS_{i,w1}} = 0.11$, $z = 1.52$, *SE* = 0.08, $p = 0.130$ and $\phi_{HPC_{w1},LS_{d,w1}} = 0.14$, $z = 1.93$, *SE* = 0.07, $p = 0.054$). We further examined whether in the reduced dataset the hippocampal contribution at delayed feedback would selectively enhance episodic memory. Episodic memory, as measured by individual corrected object recognition memory (hits – false alarms) of confident ('sure') ratings was indeed significantly enhanced for delayed feedback ($\beta_{feedback=delayed} = 0.011$, *SE* = 0.005, $t(124) = 2.23$, $p = 0.027$), which was not the case in the results when using the complete dataset.

The results obtained from the reduced dataset suggest that the striatal associations to learning remained unchanged, while the results for the hippocampus differed. The hippocampal volume was no longer associated with the delayed learning condition. Furthermore, the hippocampal-dependent episodic recognition memory was enhanced for items encoded during delayed compared to immediate feedback, which was not the case in the results obtained from the complete dataset.

