## [Editor Report · eLife assessment]

In this work, the authors make a **valuable** contribution based on **convincing** evidence that children 6-to-7-years-old improve in 2 years of development towards utilising more optimal value-based decision-making strategies while performing a reinforcement learning task. They found that delayed feedback learning was associated with volume in the hippocampus while immediate feedback learning was not. Striatal volume was associated with both forms of learning, in contrast to prior research funding in adults. Brain-behaviour correlations were stable across the 2-year period, despite the hippocampus increasing in volume and striatal volume remaining stable.

---

## [Referee Report · Reviewer #1 (Public Review)]

Existing literature suggests that brain structures implicated in memory such as the hippocampus, and reward/punishment processing such as the striatal regions are also engaged in learning and value-based decision-making. However, how the contributions of these regions to learning and value-based decision-making change over time, particularly in children where these neural systems show protracted maturation was not studied systematically. This is the question the authors are aiming to address in this work in which children 6-to-7-years-old were recruited for a neuroimaging study that involves taking structural scans from this cohort to investigate how they correlate with changes in the way children approach a reinforcement learning task in which they learn to identify the better shape between 2 options through trial-and-error.

Particular strengths of the paper are longitudinally following up a cohort of small children and engaging them in a value-based decision-making task so that the relationship between neural maturation and improvements in reinforcement learning can be studied reliably. Towards this end, the authors make use of well-established computational modelling approaches to extract key parameters such as learning rates (which designate the speed of learning from expected versus actual outcomes) or choice stochasticity (which designate the inherent variation in people's decisions and the tendency to explore between the options) from children's choices so that their structural neural correlates can be established. As a part of this endeavour, the authors rely on methodological choices which do not warrant much criticism. Their data visualization choices are particularly spot-on and highly informative about the details of the raw data.

Also considering the importance of the hippocampal system in human memory, the key contribution of the paper is that the volumetric increases in hippocampus size between 2 assessment points correlated selectively with the delayed, but not immediate, learning score which refers to the learning condition in which the outcome feedback is given to the children after a 5-seconds delay. Although the authors also demonstrate evidence to suggest that changes in the striatal volume are also implicated in learning performance, this was more general as associations were found for both immediate and delayed feedback conditions. Thus, the paper makes an important contribution to the fields of developmental and decision neuroscience. An important question arising from the authors' findings could be that, whether the hippocampus maintains this selective role in value-based learning during the course of neuronal development, for example, whether a similar association would be found in children 8-to-9 years old. A better understanding of how these developmental trajectories map onto changes in learning and decision-making can inform fields outside neuroscience, for example tailoring educational approaches onto neural development pathways to boost learning efficiency in young children.

---

## [Referee Report · Reviewer #2 (Public Review)]

Summary:

This is an interesting and impressive study that provides a rare opportunity to learn about brain-behaviour links of learning systems at a relatively early stage of development.

The main strengths are that the authors followed a relatively large group of children over 2 years and used a reinforcement learning task aimed at assessing learning that depends on both the striatum and the hippocampus. The authors also included a thorough overview of the computational models and the choices they made. I think this paper would be of considerable interest and contributes to knowledge about how learning and memory systems change with development.

---

## [Author Response]

The following is the authors’ response to the original reviews.

**Reviewer #1 (Recommendation for the authors):**
(1) On a few occasions, I found that the authors would introduce a concept, but provide evidence much later on. For example, in line 57, they introduced the idea that feedback timing modulates engagement of the hippocampus and striatum, but they provided the details much later on around line 99. There are a few instances like these, and the authors may want to go through the manuscript critically to bridge such gaps to improve the flow of reading.

First, we thank the reviewer for acknowledging the contribution of our study and the methodological choices. We acknowledge the concern raised about the flow of information in the introduction. We have critically reviewed the manuscript, especially on writing style and overall structure, to ensure a smoother transition between the introduction of concepts and the provision of supporting evidence. In the case of the concept of feedback timing and memory systems, lines 46-58 first introduce the concept enhanced with evidence regarding adults, and we then pick up the concept around line 103 again to relate it to children and their brain development to motivate our research question. To further improve readability, we have included an outline of what to expect in the introduction. Specifically, we added a sentence in line 66-68 that provides an overview of the different paragraphs: “We will introduce the key parameters in reinforcement learning and then we review the existing literature on developmental trajectories in reinforcement learning as well as on hippocampus and striatum, our two brain regions of interest.”

This should prepare the reader better when to expect more evidence regarding the concepts introduced. We included similar “road-marker” outline sentences in other occasions the reviewer commented on, to enhance consistency and readability.

(2) I am curious as to how they think the 5-second delay condition maps onto real-life examples, for example in a classroom setting feedback after 5 seconds could easily be framed as immediate feedback.The authors may want to highlight a few illustrative examples.

Thank you for asking about the practical implications of a 5-second delay condition, which may be very relevant to the reader. We have modified the introduction example in line 39-41 towards the role of feedback timing in the classroom to point out its practical relevance early on: “For example, children must learn to raise their hand before speaking during class. The teacher may reinforce this behavior immediately or with a delay, which raises the question whether feedback timing modulates their learning”.

We have also expanded a respective discussion point in lines 720-728 to pick up the classroom example and to illustrate how we think timescale differences may apply: “In scenarios such as in the classroom, a teacher may comment on a child’s behavior immediately after the action or some moments later, in par with our experimental manipulation of 1 second versus 5 seconds. Within such short range of delay in teachers’ feedback, children’s learning ability during the first years of schooling may function equally well and depend on the striatal-dependent memory system. However, we anticipate that the reliance on the hippocampus will become even more pronounced when feedback is further delayed for longer time. Children’s capacity for learning over longer timescales relies on the hippocampal-dependent memory system, which is still under development. This knowledge could help to better structure learning according to their development.”

(3) In the methods section, there are a few instances of task description discrepancies which make things a little bit confusing, for example, line 173 reward versus punishment, or reward versus null elsewhere e.g. line 229. In the same section, line 175, there are a few instances of typos.

We appreciate your attention to detail in pointing out discrepancies in task descriptions and typos in the method section. We have revised the section, corrected typos, and now phrased the learning outcomes consistently as “reward” and “punishment”.

(4). I wasn't very clear as to why the authors did not compute choice switch probability directly from raw data but implemented this as a model that makes use of a weight parameter. Former would-be much easier and straightforward for data plotting especially for uninformed readers, i.e., people who do not have backgrounds in computational modelling.

Thank you for asking for clarification on the calculation of switching behavior. Indeed, in the behavioral results, switching behavior was directly calculated from the raw data. We now stressed this in the methods in lines 230-235, also by naming win-stay and lose-shift as “proportions” instead of as “probabilities”:“As a first step, we calculated learning outcomes diretly from the raw data, which where learning accuracy, win-stay and lose-shift behavior as well as reaction time.

Learning accuracy was defined as the proportion to choose the more rewarding option, while win-stay and lose-shift refer to the proportion of staying with the previously chosen option after a reward and switching to the alternative choice after receiving a punishment, respectively.”

In contrast to the raw data switching behavior, the computational heuristic strategy model indeed uses a weight for a relative tendency of switching behavior. We have also stressed the advantage of the computational measure and its difference to the raw data switching behavior in lines 248-252 and believe that the reader can now clearly distinguish between the raw data and the computational results: “Note that these model-based outcomes are not identical to the win-stay and lose-shift behavior that were calculated from the raw data. The use of such model-based measure offers the advantage in discerning the underlying hidden cognitive process with greather nuance, in contrast to classical approaches that directly use raw behavioral data.”

(5) I agree with the authors' assertion that both inverse temperature and outcome sensitivity parameters may lead to non-identifiability issues, but I was not 100% convinced about their modelling approach exclusively assessing a different family of models (inv temperature versus outcome sensitivity). Here, I would like to make one mid-way recommendation. They may want to redefine the inverse temperature term in terms of reaction time, i.e., B=exp^(s+g(RT-mean (RT))) where s and g are free parameters (see Webb, 2019), and keep the outcome sensitivity parameter in the model with bounds [0,2] so that the interpretation could be % increase or decrease in actual outcome. Personally, in tasks with binary outcomes i.e. [0,1: null vs reward] I do not think outcome sensitivity parameters higher than 2 are interpretable as these assign an inflated coefficient to outcomes.

We appreciate the mid-way recommendation regarding the modeling approach for inverse temperature and outcome sensitivity parameters. We have carefully revised our analysis approach by considering alternative modeling choices. Regarding the suggestion to redefine the inverse temperature in terms of reaction time by B=exp^(s+g(RT-mean (RT))), we unfortunately were not able to identify the reference Webb (2019), nor did we find references to the suggested modeling approach. Any further information that the reviewer could provide will be greatly appreciated. Regardless, we agree that including reaction times through the implementation of drift-diffusion modeling may be beneficial. However, changing the inverse temperature model in such a way would necessitate major changes in our modeling approach, which unfortunately would result in non-convergence issues in our MCMC pipeline using Rstan. Hence, this approach goes beyond the scope of the manuscript. Nonetheless, we have decided to mention the use of a drift-diffusion model, along with other methodological considerations, as future recommendation for disentangling outcome sensitivity from inverse temperature in lines 711-712: “Future studies might shed new light by examining neural activations at both task phases, by additionally modeling reaction times using a drift-diffusion approach, or by choosing a task design that allows independent manipulations of these phases and associated model parameters, e.g., by using different reward magnitudes during reinforcement learning, or by studying outcome sensitivity without decisionmaking.“

Regarding the upper bound of outcome sensitivity, we agree that traditionally, limiting the parameter values at 2 is the choice for the parameter to be best interpretable. During model fitting, we had experienced non-convergence issues and ceiling effects in the outcome sensitivity parameter when fixing the inverse temperature at 1. The non-convergence issue was not resolved when we fixed the inverse temperature at 15.47, which was the group mean of the winning inverse temperature family. Model convergence was only achieved after increasing the outcome sensitivity upper bound to 20, with inverse temperature again fixed at 1. Since this model also performed well during parameter and model recovery, we argue that the parameter is nevertheless meaningful, despite the more extreme trial-to-trial value fluctuations under higher outcome sensitivity. We described our choice for this model in the methods section in lines 282-288: “Even though outcome sensitivity is usually restricted to an upper bound of 2 to not inflate outcomes at value update, this configuration led to ceiling effects in outcome sensitivity and non-converging model results. Further, this issue was not resolved when we fixed the inverse temperature at the group mean of 15.47 of the winning inverse temperature family model. It may be that in children, individual differences in outcome sensitivity are more pronounced, leading to more extreme values. Therefore, we decided to extend the upper bound to 20, parallel to the inverse temperature, and all our models converged with Rhat < 1.1.”.

(6) I think the authors reporting optimal parameters for the model is very important (line 464), but the learning rate they report under stable contingencies is much higher than LRs reported by for example Behrens et al 2007, LRs around 0.08 for the optimal learning behaviour. The authors may want to discuss why their task design calls for higher learning rates.

Thank you for appreciating our optimal parameter analysis, and for the recommendation to discuss why optimal learning rates in our task design may call for higher learning rates compared to those reported in some other studies. As largely articulated in Zhang et al (2020; primer piece by one of our co-authors), the optimal parameter combination is determined by several factors, such as the reward schedule (e.g., 75:25, vs 80:20) and task design (e.g., no reversal, one reversal, vs multiple reversal) and number of trials (e.g., 80, vs 100, vs, 120). Notably, in these taskrelated regards, our task is different from Behrens et al. (2007), which hinders a quantitative comparison among the optimal parameters in the two tasks. We have now included more details in our discussion in lines 643-656: “However, the differences in learning rate across studies have to be interpreted with caution. The differences in the task and the analysis approach may limit their comparability. Task proporties such as the trial number per condition differed across studies. Our study included 32 trials per cue in each condition, while in adult studies, the trials per condition ranged from 28 to 100. Optimal learning rates in a stable learning environment were at around 0.25 for 10 to 30 trials, another study reported a lower optimal learning rate of around 0.08 for 120 trials. This may partly explain why in our case of 32 trials per condition and cue, optimal learning rates called for a relatively high optimal learning rate of 0.29, while in other studies, optimal learning rates may be lower. Regarding differences in the analysis approach, the hierarchical bayesian estimation approach used in our study produces more reliable results in comparison to maximum likelihood estimation, which had been used in some of the previous adult studies and may have led to biased results towards extreme values. Taken together, our study underscores the importance of using longitudinal data to examine developmental change as well as the importance of simulation-based optimal parameters to interpret the direction of developmental change.”

(7) The authors may want to report degrees of freedom in t-tests so that it would be possible to infer the final sample size for a specific analysis, for example, line 546.

We appreciate the recommendation to include degrees of freedom, which are now added in all t-test results, for example in line 579: “Episodic memory, as measured by individual corrected object recognition memory (hits - false alarms) of confident (“sure”) ratings, showed at trend better memory for items shown in the delayed feedback condition (𝛽!""#$%&’(#")%*"# = .009, SE = .005, t(137) = 1.80, p = .074, see Figure 5A).”

(8) I'm not sure why reductions in lose shift behaviour are framed as an improvement between 2 assessment points, e.g. line 578. It all depends on the strength of the contingency so a discussion around this point should be expanded.

We acknowledge that a reduction in lose-shift behavior only reflect improvements under certain conditions where uncertainty is low and the learning contingencies are stable, which is the case in our task. We have added Supplementary Material 4 to illustrate the optimality of win-stay and lose-shift proportions from model simulation and to confirm that children’s longitudinal development was indeed towards more optimal switching behavior. In the manuscript, we refer to these results in lines 488-490: “We further found that the average longitudinal change in win-stay and lose-shift proportion also developed towards more optimal value-based learning (Supplementary Material 4).”

(9) If I'm not mistaken, the authors reframe a trend-level association as weak evidence. I do not think this is an accurate framing considering the association is strictly non-significant, therefore should be omitted line 585.

We thank for the point regarding the interpretation of a trend-level association as weak evidence. We changed our interpretation, corrected in lines 581-585: “The inclusion of poor learners in the complete dataset may have weakend this effect because their hippocampal function was worse and was not involved in learning (nor encoding), regardless of feedback timing. To summarize, there was inconclusive support for enhanced episodic memory during delayed compared to immediate feedback, calling for future study to test the postulation of a selective association between hippocampal volume and delayed feedback learning.” as well as lines 622-623: “Contrary to our expectations, episodic memory performance was not enhanced under delayed feedback compared to immediate feedback.”

**Reviewer # 2 (Public Review):**

We thank the reviewer for acknowledging the strength of our study and pointing out its weaknesses.

Weaknesses:There were a few things that I thought would be helpful to clarify. First, what exactly are the anatomical regions included in the striatum here?

We appreciate the clarification question regarding the anatomical regions included in the striatum. The striatum included ventral and dorsal regions, i.e., accumbens, caudate and putamen. We have now specified the anatomical regions that were included in the striatum in lines 211-212: “We extracted the bilateral brain volumes for our regions of interest, which were striatum and hippocampus. The striatum regions included nucleus accumbens, caudate and putamen.”

Second, it was mentioned that for the reduced dataset, object recognition memory focused on "sure" ratings. This seems like the appropriate way to do it, but it was not clear whether this was also the case for the full analyses in the main text.

Thank you for pointing out that in the full dataset analysis, the use of “sure” ratings for object recognition memory was previously not mentioned. Including only “sure” ratings was used consistently across analyses. This detail is now described under methods in lines 332-333: “Only confident (“sure”) ratings were included in the analysis, which were 98.1 % of all given responses.”

Third, the children's fitted parameters were far from optimal; is it known whether adults would be closer to optimal on the task?

We thank for your question on whether adult learning rates in the task have been reported to be more optimal than those of the children in our study. This indeed seems to be the case, and we added this point in our discussion in line 639-643: “Adult studies that examined feedback timing during reinforcement learning reported average learning rates range from 0.12 to 0.34, which are much closer to the simulated optimal learning rates of 0.29 than children’s average learning rates of 0.02 and 0.05 at wave 1 and 2 in our study. Therefore, it is likely that individuals approach adult-like optimal learning rates later during adolescence.”

The main thing I would find helpful is to better integrate the differences between the main results reported and the many additional results reported in the supplement, for example from the reduced dataset when excluding non-learners. I found it a bit challenging to keep track of all the differences with all the analyses and parameters. It might be helpful to report some results in tables side-by-side in the two different samples. And if relevant, discuss the differences or their implication in the Discussion. For example, if the patterns change when excluding the poor learners, in particular for the associations between delayed feedback and hippocampal volume, and those participants were also those less well fit by the value-based model, is that something to be concerned about and does that affect any interpretations? What was not clear to me is whether excluding the poor learners at one extreme simply weakens the general pattern, or whether there is a more qualitative difference between learners and non-learners. The discussion points to the relevance of deficits in hippocampaldependent learning for psychopathology and understanding such a distinction may be relevant.

We appreciate the feedback that it might seem challenging to keep track of differences between the analyses of the full and the reduced dataset. We have now gathered all the analyses for the reduced dataset in Supplementary Material 6, with side-by-side tables for comparison to the full dataset results. Whenever there were differences between the results, they were pointed out in the results section, see lines 557-560: “In the results of the reduced dataset, the hippocampal association to the delayed learning score was no longer significant, suggesting a weakened pattern when excluding poor learners (Supplementary Material 6). It is likely that the exclusion reduced the group variance for hippocampal volume and delayed learning score in the model.” and lines 579-581: “Note that in the reduced dataset, delayed feedback predicted enhanced item memory significantly (Supplementary Material 6).”

The found differences were further included in our discussion in lines 737-740 in the context of deficits in hippocampal-dependent learning and psychopathology: “Interestingly, poor learners showed relatively less value-based learning in favor of stronger simple heuristic strategies, and excluding them modulated the hippocampal-dependent associations to learning and memory in our results. More studies are needed to further clarify the relationship between hippocampus and psychopathology during cognitive and brain development.”